# Animal- and Plant-Based Protein Sources: A Scoping Review of Human Health Outcomes and Environmental Impact

**DOI:** 10.3390/nu14235115

**Published:** 2022-12-01

**Authors:** Luca Ferrari, Stefan-Alexandru Panaite, Antonella Bertazzo, Francesco Visioli

**Affiliations:** 1Department of Molecular Medicine, University of Padova, 35122 Padova, Italy; 2Department of Cardiac, Thoracic, Vascular Sciences, and Public Health, School of Hygiene and Preventive Medicine, University of Padova, 35122 Padova, Italy; 3Department of Pharmaceutical and Pharmacological Sciences, University of Padova, 35122 Padova, Italy; 4IMDEA-Food, CEI UAM+CSIC, 28001 Madrid, Spain

**Keywords:** protein sources, protein quality, main chronic diseases, vegetarian diets, environmental impact, climate change

## Abstract

Dietary proteins are indispensable to human nutrition. In addition to their tissue-building function, they affect body composition and regulate various metabolic pathways, as well as satiety and immune system activity. Protein use can be examined from a quantitative or qualitative viewpoint. In this scoping review, we compare animal- and plant-based protein sources in terms of their effects on human health and the environment. We conclude that the consumption of vegetable protein sources is associated with better health outcomes overall (namely, on the cardiovascular system) than animal-based product use. The healthier outcomes of vegetable protein sources dovetail with their lower environmental impact, which must be considered when designing an optimal diet. Indeed, the health of the planet cannot be disjointed from the health of the human being. Future research will clarify the mechanisms of action underlying the health effects of plant-based protein sources when compared with animal sources, fostering better agronomic practices and influencing public health in a direction that will benefit both the planet and its inhabitants.

## 1. Introduction

Dietary proteins are indispensable to human nutrition. Their most important biological function is tissue-building. However, proteins also affect body composition and regulate various metabolic pathways, as well as satiety and immune system activity [1,2].

In the field of nutrition, protein consumption can be examined from a quantitative or qualitative viewpoint. Regarding the former, a recommended daily allowance (RDA) has been set at 0.8 g/kg of body weight (bw) for healthy adults under normal conditions [3], but this varies depending on some factors, e.g., age, some physiological needs, pregnancy, etc. For example, athletes should likely increase their RDA to 1.2–1.6 g/kg of bw [3].

In terms of their qualitative profile, it is important to analyze protein quality, i.e., a parameter that categorizes proteins into two groups on the basis of their amino acid composition: high- and low-quality proteins. In lay terms, this distinction is often referred to as animal- and plant-based proteins, respectively.

Another important factor that modulates the contribution of proteins to human health is the overall profile of protein-containing foods: the Acceptable Macronutrient Distribution Range (AMDR) considers the optimal range of protein intake to be between 10 and 35% of energy intake. Excluding dried edible insects, which can contain up to 60 g of protein per 100 g of product [4], and some other sparsely consumed foods, such as soy protein isolate, stockfish, scraps of swine fat, and roe, the most commonly consumed foods rich in protein reach a maximum protein content of between 33 g and 37 g per 100 g of product. Some notable examples are dried soya and the flour derived from it, Parmesan cheese, cured meat, and guinea fowl [5]. For this reason, it should be more appropriate to talk about “protein sources” rather than “proteins”.

High-income countries currently consume too many animal-based products and too few plant-based ones. This imbalance is thought to contribute to an increase in the risk of chronic diseases [6].

For this reason, nutritional guidelines emphasize the need to shift toward plant-based diets (though not necessarily vegetarian diets). These recommendations dovetail with the excessive pressure that animal-based food systems exert on the planet. The syndemic relationship between chronic diseases and climate change is the subject of active research [7] and is crucial to understanding the overall effects of the consumption of protein sources on human health and the environment.

In this scoping review, we analyzed the main food groups classified as protein sources [8] and segregated them into the two macro-categories of “animal protein sources” and “plant-based protein sources”.

This analysis was carried out with priority given to the assessment of the direct effects of such foods on human health; consequently, correlations between their consumption and the most common chronic diseases were evaluated.

In addition, the environmental impacts of these foods have also been investigated with the aim of returning a more complete picture of the implications of different protein sources.

In evaluating this intricate picture, a brief but punctual description of some topics related to protein sources, such as protein quality, vegetarian and plant-based dietary patterns, and the double food pyramid of the Mediterranean Diet, is also reported.

## 2. Materials and Methods

We searched the most popular databases (PubMed, Google Scholar, WebScience, Cochrane Library, and Scopus) for the most recent publications and reviews on this topic, using “plant-based protein” or “plant-based protein sources”, “vegetable protein” or “vegetable protein sources”, and “animal protein” or “animal protein sources” as keywords. We extracted the resulting publications, which were read, interpreted, and discussed among the authors.

For the topic of protein quality and the double food pyramid, we drew from various narrative reviews, some consensus papers, and systematic reviews; for the analysis of the vegetarian and Mediterranean dietary patterns, we used studies similar to the previous ones, as well as meta-analyses of prospective cohort studies and umbrella reviews, to assess the association between these two dietary patterns and chronic diseases. The latter types of papers were used for the creation of tables on individual protein sources, whereas interventional studies (i.e., randomized controlled trials, RCTs) were not considered.

We built tables as follows: the dark-gray background stands for a lack of studies (no data); the gray background (with a slash included in cells) indicates an absence of significance; light gray indicates a “nearly significant” association (inverse or positive); all other data with a white background illustrate statistically significant associations (inverse or positive).

## 3. Dietary Protein Quality

To compare animal and plant-based protein sources from the viewpoint of both nutrition and sustainability, it is useful to start from the concept of “protein quality” [9]. There is a current debate on the need to revise and broaden this notion: in lay terms, “quality” is almost always associated with the idea of “desirability”, and therefore, its use in this context may be inappropriate and lead to misinterpretations [10].

In describing protein quality, different approaches may be taken. The most traditional approach is to consider the positive biochemical impact of dietary proteins on protein synthesis and nitrogen balance [11], An alternative approach is to consider the impact of dietary proteins on the function and metabolism of specific organs and/or hormones (e.g., evaluating the regulation of body composition and bone health, gastrointestinal (GI) function and bacterial flora, glucose homeostasis, cell signaling, satiety) [12]. The most recent and far-reaching approach is to consider the net effects of diet on human health and the environment [10].

### 3.1. Traditional Approach to Protein Quality Evaluation

In the traditional approach, the quality of a given protein source can be defined as its ability to efficiently meet the requirements of both nitrogen and essential amino acids (EAAs), and it depends on two factors:the specific amino acid composition (the intrinsic quality of proteins);digestibility (the extrinsic quality of proteins) [13].

Concerning the first component, EAAs are a key factor in determining the intrinsic quality of proteins, which is represented by the combination of the nine EAAs’ content.

If the dietary content of a single EAA is lower than the individual’s reference requirement, this specific deficiency limits the use of any other amino acid. Therefore, even if the total nitrogen intake level is adequate, the “limiting amino acid” prevents normal rates of protein synthesis and determines the nutritional value of the total protein (or nitrogen) content in the diet [13]. This is, however, a theoretical and academic simplification, given that the evaluation of the daily diet is more complex, because it contains a mixture of different protein sources. It is possible to circumvent the “limiting amino acid” problem through the precise complementation of protein sources [14] (see below).

The concept of “limiting amino acid” has led to the development of amino acid scoring and, consequently, to a reference amino acid scoring pattern. Although there are currently some uncertainties in establishing this system, it is useful for comparing the quality of different protein sources [11,13].

For extrinsic protein quality, there are two processes to consider:the digestion of proteins and the absorption of the constituent amino acids (so-called digestibility) [9]. «Digestibility is defined as the difference between the amount of N ingested and excreted, expressed as a proportion of N ingested». Due to the processes of protein metabolization of the intestinal microbiota, it is more appropriate to consider ileal digestibility than the fecal digestibility [15]. More precisely, it is necessary to measure the true ileal digestibility (TID), which also takes into account the endogenous protein losses (both basal and specific ones) [15].The utilization of the absorbed amino acids to support whole-body protein synthesis (so-called availability).

These two different processes constitute what can be defined as the bioavailability (or metabolic availability) of nutrients, in this specific case, that of amino acids from dietary protein sources. In 1991, a score was proposed to merge these two components and then to evaluate the protein quality quantitatively [16]; this is the Protein Digestibility Corrected Amino Acid Score (PDCAAS, reference proteins are e.g., egg white or casein), whose formula is:$$PDCAAS\;\left(\%\right)=\frac{\left[mg\;of\;limiting\;amino\;acid\;in\;1\;g\;test\;protein\right]}{mg\;of\;same\;amino\;acid\;in\;1\;g\;reference\;protein}\;\times \;[true\;digestibility\;(D_{F})(\%)]$$

This score has some limits, such as: (1) the truncation of the score to the value of 100%, which does not allow for a comparison between proteins with very high scores [17]; (2) the use of fecal digestibility rather than TID [17]; (3) the restriction to only the first limiting amino acid [17]; (4) the lack of consideration of the bioavailability of every single EAA [12,18].

The latter issue is particularly important, since the relevance of considering amino acids as individual nutrients has recently been understood [19].

In order to augment the PDCAAS, in 2011, the “FAO Expert Consultation on Protein Quality Evaluation in Human Nutrition” introduced another score, the Digestible Indispensable Amino Acid Score (DIAAS). DIAAS is calculated as follows [18]:$$DIAAS\;\left(\%\right)=\frac{100\;x\left[mg\;of\;digestible\;dietary\;IAA\;in\;1\;g\;test\;protein\right]}{mg\;of\;same\;amino\;acid\;in\;1\;g\;reference\;protein}\;\times \;[true\;digestibility\;(D_{F})(\%)]$$

According to this score, protein sources are classified into three categories [18]: (1) “excellent” sources (DIAAS > 100%); (2) “good” sources (75% < DIAAS < 100%); and (3) “no claim” sources (DIAAS < 75%).

Because they generally contain reduced amounts of one or more EAAs, plant-based protein sources have lower DIAAS values (and, for that matter, PDCAAS values) and are therefore defined as “low-quality protein sources” [20].

The term “incomplete protein sources” «is no longer commonly used as all protein sources contain all 20 amino acids» [21].

However, plant-based protein sources are complementary to each other. The classic example of protein complementation is the combination of cereals and legumes (for simplicity, think of the Italian recipe of pasta and beans): legumes typically have low/limiting levels of methionine (a sulfur-containing amino acid) and high levels of lysine. Cereals have exactly the opposite characteristics [20].

### 3.2. Recent Approaches to Protein Quality Evaluation

Because dietary proteins have more biological functions than the mere maintenance of body protein mass, the concept of protein quality has to be broadened in order to include new elements in a quantitative evaluation. To achieve this new “protein quality metric”, it is also necessary to consider intake levels that are higher than the RDAs as normal, at least in high-income countries [12]. Such levels were established based only on the maintenance of the nitrogen balance in nearly all (97.5%) healthy individuals [22]. At the same time, it is also necessary to consider the environmental impact of protein use. Of all the various factors that can be measured through the footprint methodology [23], the main ones related to food systems could be conveniently classified as: (a) greenhouse gas (GHG) emissions (carbon footprint); (b) freshwater (or blue water) use (water footprint); (c) cropland use (land footprint or the wider ecological footprint); (d) cumulative energy demand (CED, carbon component of the ecological footprint); (e) the use of chemicals, such as fertilizers and pesticides (specific component of material footprint for fertilizers and specific component of chemical footprint for pesticides); (f) loss of biodiversity (biodiversity footprint).

A first attempt to harmonize such measurements was proposed by Katz et al. [10] through the definition of two different sample metrics.

Yet, given the enormous complexity of the topic in terms of both nutrition and sustainability, multiple aspects must be evaluated [10,14]. Currently, there are various approaches as to how the different factors should be considered. A pertinent example is provided by an Italian study, which assessed how the cropland use and GHG emissions of various protein sources should be measured [24]. The urgency of developing an evidence-based tool in this specific area is well-highlighted in a rather recent article [25], in which the Life Cycle Assessment (LCA) [26,27,28] is proposed as the benchmark method of analysis.

## 4. Dietary Patterns and Protein Intake

The various dietary patterns can be conveniently differentiated according to various methods [26,29]. An a priori index is derived from a series of dietary recommendations for a healthy and high-quality dietary model. The second method of dietary pattern assessment consists of “data-driven” approaches, such as cluster analysis and factor analysis, that are outcome-independent and derive a posteriori patterns.

The third method is based on individual preferences in food and beverage intake.

### 4.1. Vegetarian Dietary Patterns

Vegetarian dietary patterns are an emblematic example of the third assessment method, but they can also be characterized by the first a priori method through a more generic distinction in “plant-based diets”.

The various vegetarian dietary patterns differ based on their protein sources: from the semi-vegetarian diet (flexitarian) to the vegan one, animal meat (red meat, poultry, and seafood) and animal derivatives (eggs and dairy products) are progressively eliminated [30,31]. Another slightly different definition of vegetarian patterns is based on the frequency (greater or less than one time per month) of the intake of different animal protein sources [32]. However, several studies have shown that the differences in the intake of the various foods between the omnivorous diet and the various subtypes of vegetarian diets do not only concern animal protein sources.

In the cohorts of the Adventist Health Study 2 (AHS-2) [33,34] and the European Prospective Investigation into Cancer and Nutrition (EPIC) (Oxford) [35]—the two largest of the known ongoing prospective studies that include high proportions of vegetarians—and that of the National Health and Nutrition Examination Surveys (NHANES) 2013–2016 [36], vegetarian dietary patterns are characterized by: (a) a greater intake of certain major food categories (clearly plant-based protein sources (legumes, seeds, and nuts) but also fruits, vegetables, whole grains and tubers) and of some micronutrients (fibers, minerals (magnesium, above all), and vitamins (A, C, and E types)) [37]; (b) a lower intake of other food categories (refined cereals, saturated fatty acids (SFAs) of animal origin, added fats, sweets, snack foods, and non-water beverages) and of other micronutrients (cholesterol, sodium, vitamin B12, and zinc). Therefore, the differences in terms of health outcomes (see below) cannot be directly linked only to the different consumption of protein sources.

#### 4.1.1. Health Outcomes

The opening statement of the first edition of the “Dietary Reference Intakes for Energy, Carbohydrate, Fiber, Fat, Fatty Acids, Cholesterol, Protein, and Amino Acids” reported that «a varied vegetarian diet can get the same quality of protein and similar amounts of nitrogen as yielded by animal protein or a mixed diet» [38]. Moreover, the US scientific report of the 2015 Dietary Guidelines Advisory Committee (DGAC) also stated that a “Healthy Vegetarian Pattern” is associated with health benefits [26]. In addition, the Italian Society of Human Nutrition declared through a position paper that «well-planned vegetarian diets […] provide adequate nutrient intake» [31]. A recent review has definitively dispelled the myth of the inadequacy of various vegetarian diets for protein and amino acids requirements [39].

Much research has been performed to better understand the differences in health outcomes between omnivorous diets and vegetarian diets. Among the publications we selected, there are some reviews, systematic reviews, and meta-analyses [40,41,42,43,44,45,46,47], a position paper of the US Academy of Nutrition and Dietetics [30], the US scientific report of the 2020 DGAC [48], as well as some relevant cross-sectional analyses of the AHS-2 cohort [32,34,49,50,51,52,53,54,55,56], the EPIC-Oxford cohort [56,57,58,59,60,61,62,63], and some other cohorts [63,64,65,66,67,68]. Some of the chosen papers only distinguish between vegetarians and non-vegetarians, but most subdivide the population into the four subtypes of vegetarian dietary patterns mentioned above and differently defined as:“semi-vegetarian diet” or “low-occasional meat-eaters” (flexitarian);“pesco-vegetarian diet” or “fish-eaters” (pescatarian);“lacto-ovo vegetarian diet” or more generically “vegetarians”;“vegan diet” or “vegans”.

In some of the papers we analyzed (Table 1), the all-cause mortality of vegetarian diets is comparable to or lower than that of an omnivorous diet, considering a population group with a similar lifestyle and adjusting the results for age, sex, smoking status, and alcohol consumption. Among the various analyses of prospective studies, a paper from 1999 already identified that the total risk of mortality was 5% lower in vegetarians than non-vegetarians. Although this result was not statistically significant, the data were quite reliable, because they were only staggered by the results of the vegan pattern, the less well-represented group. The other three patterns, “occasional meat-eaters”, “fish eaters”, and “vegetarians”, had a statistically significant reduced risk by 16%, 18%, and 16%, respectively [64].

Subsequent analyses have confirmed this trend among vegetarians and non-vegetarians. Comparable values in all-cause mortality were found when analyzing the results from the EPIC-Oxford cohort [57]. A meta-analysis from 2012 found a non-statistically significant decreased mortality of 9% [42], and a study from 2013 (and its update from the following year) based on the results of the AHS-2 cohort set it at 12% (statistically significant) [32,50].

Therefore, at the moment, the data on all-cause mortality drawn from the two large cohorts are contrasting. The difference between the results of the British (EPIC-Oxford) and US (AHS-2) cohort studies’ analyses might be explained by the fact that «British vegetarians and US Adventist vegetarians eat somewhat differently» [32].

To clarify this aspect, and thus to obtain wide statistical significance, other well-designed prospective studies will certainly be necessary; they should provide for the distinction into the five subgroups of vegetarian patterns and not only for the simple “vegetarian and non-vegetarian” binarism.

In any case, specific studies on longevity identify the pescatarian diet as the best option for an extended lifespan and health span [69]. In addition, two other analyses of prospective studies that used the first method of dietary pattern assessment (that is, by defining a priori indices for diets) and that were conducted in populations larger than those previously mentioned (not only Seventh-Day Adventists, vegetarians, and more general health-conscious individuals) have highlighted how plant-based diets are associated with a statistically significant decrease in all-cause mortality compared to more animal-based diets; the respective values were 24% [68] and 5% [67].

The non-vegetarian pattern and the vegetarian patterns were also compared for other health outcomes and not only for all-cause mortality. These included mortality and morbidity (incidence or prevalence) from specific groups of diseases, such as cardiovascular or metabolic diseases and cancers; and mortality and morbidity (incidence or prevalence) from a single disease, such as coronary heart disease (CHD), stroke, type 2 diabetes (T2D), or a specific cancer.

Recently, this field of research has expanded, and among many studies and many health outcomes examined here, we have highlighted those considered to be the most relevant, taking as a principal reference the recent wide umbrella review by Oussalah et al. [44].

##### Cardiovascular and Metabolic Diseases

In the field of cardiovascular diseases (CVDs), the most evidence can be drawn for CHD: several studies have shown a reduction in both mortality and incidence associated with vegetarian diets compared to a non-vegetarian diet. A statistically significant reduced mortality ranging between 24% and 29% was demonstrated [41,42,64] (accompanied by similar values that were not statistically significant from other studies [32,57]) along with a reduced incidence of between 25% and 32%, which was also statistically significant [41,59]. An even higher value of this specific decreased mortality was found in the cohort of AHS-2, more precisely associated with the pescatarian diet, with a reduction of 35% [32]. On the other hand, to date, a statistical significance between vegetarian diets (both total and subtypes) and a reduction in mortality and morbidity in cerebrovascular diseases has not been determined. Indeed, some results point to the trend of a positive outcome in the associated mortality (reduction of between 7% and 12% [41,42,64]). Other studies, conversely, have come to the opposite conclusion (an increase in mortality of 10% or greater [57,65]). In brief, vegetarian diets cannot be associated with a reduction in mortality and morbidity from CVDs. In addition to the aforementioned studies [32,41,42,57], only one paper that defined a healthy plant-based diet a priori reported a reduction in the incidence and mortality of CVDs in adults [68].

Regarding cardiovascular risk factors, vegetarian dietary patterns have been demonstrated to have overall advantages over an omnivorous diet in terms of surrogate markers. The blood pressure of subjects who followed a vegetarian diet was reduced compared to those of non-vegetarian subjects. A meta-analysis that analyzed seven clinical trials and 32 observational studies reported a statistically significant drop in both systolic pressure (about 5 mmHg) and diastolic pressure (about 2 mmHg) in the first type of study and statistically significant lower values both in systolic pressure (about 7 mmHg) and diastolic pressure (about 5 mmHg) also in the second type [46]. The analysis of the AHS-2 cohort also found that lacto-ovo vegetarian and vegan diets were associated with a significant reduction in the prevalence of high blood pressure (HBP), by 43% and 63%, respectively (in non-Black subjects) [51], and for the two diets together by 44% (in Black subjects) [52].

Vegetarians generally had a significant reduction in total blood cholesterol and LDL cholesterol. For the former, the reduction was between 28 and 29 mg/dL according to two meta-analyses of observational studies [41,47], whereas LDL cholesterol was reduced by a value between 21 and 23 mg/dL in the same two studies [41,47] and by about 0.45 mmol/L in an analysis of the EPIC-Oxford cohort [59]. Another cross-sectional analysis of the same cohort showed that LDL cholesterol values were progressively reduced in the different subtypes of vegetarian patterns. From non-vegetarian to vegan diets, these results were: 4.11 mmol/L for non-vegetarians, missing data for flexitarians, 3.85 mmol/L for pescatarians, 3.71 mmol/L for vegetarians, and 3.26 mmol/L for vegans [61].

The results of three meta-analyses also reported significantly lower HDL-cholesterol values among cross-sectional studies (between 2.7 and 3.6 mg/dL) [41,47] and a significant reduction in the analysis of controlled clinical trials (3.4 mg/dL [47] and 0.10 mmol/L [45]).

Current data on triglyceride levels associated with different types of diets are conflicting [41,45,47].

Regarding obese and overweight individuals, there are significant differences between vegetarians and omnivores in both prevalence and incidence. The EPIC-Oxford cohort analysis found significant discrepancies between various dietary patterns in terms of the average value of body mass index (BMI) adjusted for lifestyle factors (24.39 kg/m^2^ meat-eaters, 23.35 kg/m^2^ pescatarian, 23.38 kg/m^2^ vegetarian, 22.53 kg/m^2^ vegan; these data refer to males, but similar differences were also present in females) [62]. Different yet equally progressive lower values from the omnivorous diet to the vegan diet were found in the AHS-2 cohort (28.3 kg/m^2^ non-vegetarian, 27.3 kg/m^2^ flexitarian, 26.0 kg/m^2^ pescatarian, 26.1 kg/m^2^ vegetarian, 24.1 kg/m^2^ vegan; sex-adjusted data) [32]. These data were confirmed by the recent meta-analysis by Dinu et al. [41]: the average BMI of vegetarians was significantly lower than that of omnivores, with a weighted mean difference (WMD) of −1.49 kg/m^2^. A 2005 cross-sectional study of the Swedish Mammography Cohort highlighted a different prevalence of overweight and obese individuals among various patterns (40% non-vegetarian, 29% flexitarian, missing data pescatarian, 25% vegetarian, 29% vegan). Flexitarian, vegetarian, and vegan women also presented a significantly lower risk than non-vegetarian women in terms of incidence of these two conditions, by 48%, 46%, and 65%, respectively [66].

Significant differences in the prevalence and incidence of T2D were highlighted in the two cohorts of the AHS-2 analyses. The first analysis focused on different prevalence in various dietary patterns (7.6% non-vegetarian, 6.1% flexitarian, 4.8% pescatarian, 3.2% vegetarian, and 2.9% vegan), with a reduced risk for vegetarian patterns of 24%, 30%, 46%, and 49%, respectively [53]. The second analysis found a lower incidence for vegetarian patterns of 51% (flexitarian), 21% (pescatarian, not statistically significant), 38% (vegetarian), and 62% (vegan) [54]. Data from a recent meta-analysis confirmed this trend, setting the significant reduction in incidence to 28.4% overall for vegetarian diets [43]. These data were also consistent with the significantly lower blood glucose level among vegetarians reported by the paper of Dinu et al. [41]. The main analysis of the AHS-2 cohort study also suggested an association between vegetarian diets and reduced diabetes-related mortality [32].

Regarding metabolic syndrome (MetS), a reduced prevalence (39.6% non-vegetarian, 37.6% flexitarian and pescatarian considered together, 25.2% vegetarian and vegan together as well) and a decreased (56% for vegetarian and vegan diets) incidence of this clinical condition have been highlighted [55].

##### Cancers

Many studies have focused on the link between cancer and different vegetarian dietary patterns. Nonetheless, the results were not decisive, and thus clinically useful conclusions cannot be drawn. Major differences are visible for global cancer incidence, since two relevant distinct studies have identified a significant reduction in vegetarian patterns considered overall: 8% in the analysis of the AHS-2 cohort [49] and 18% in the meta-analysis by Huang et al. [42]. The recent meta-analysis taken as a reference [41] confirmed these data and reported an 8% reduction for incidence.

The EPIC-Oxford cohort calculated this outcome only considering separately the various patterns and reported that there was a significant reduction for pescatarians (12%), vegetarians (11%), and vegans (19%) [63]. Up to now, however, no study has shown a decreased cancer-related mortality associated with vegetarian dietary patterns, and there are inconclusive or contrasting data in cancer-specific morbidity.

#### 4.1.2. Environmental Impact

In the analysis of vegetarian and non-vegetarian dietary patterns, a fundamental element to be evaluated is unequivocally their environmental impact: adopting a vision of planetary health and considering the trilemma “diet–environment–health” [70,71] is necessary in a historical phase of demographic increase and given the remarkable influence of food systems on the current climate crisis and environmental degradation (globally responsible for 21–37% of GHG emissions and 70% of freshwater use) [6,72,73,74].

A relevant study by Springmann et al. [75] highlighted the important role of dietary change in reducing the impact of all four considered elements of “environmental pressure”—carbon, water, and land footprints and fertilizer use—especially in the predictions made for 2050, in which it is stated that the impact of the food systems could increase between 50% and 90% over the next 30 years [75].

In this study, current dietary habits were analyzed against the actual implementation of nutritional guidelines and the shift towards a flexitarian diet (“more plant-based” diet). The authors reported that the latter element is particularly relevant in reducing one’s carbon footprint and is necessary in order to remain definitively below the specific planetary boundaries, despite the simultaneous achievements of maximum improvements in technology and the maximum management of and reduction in food loss and waste.

To a lesser extent, the transition to a flexitarian diet also plays an important role in the other three environmental impact factors (water footprint, cropland use, and fertilizer use, divided into nitrogen (N footprint) and phosphorus (P footprint) application) [75].

A parallel paper by some of the authors of the previous study [76] also analyzed the reduction in premature mortality and in the environmental impact factors seen by considering not only a flexitarian diet, but different dietary patterns—called diet scenarios for the research method used—that were subdivided into three different groups. The first included plant-based diets based on environmental objectives that progressively replaced (from 25% (ani-25) to 100% (ani-100)) animal-source foods with isocaloric plant-based ones (fixed mix of 75% legumes and 25% fruit-vegetables). The second group included patterns that were not relevant to this paper (based on food security objectives and which improved energy imbalances by 25–50–75–100% and simultaneously reduced levels of underweight, overweight, and obese individuals). The third group included vegetarian dietary patterns “based on public health objectives” (vegan, vegetarian, pescatarian, and precisely flexitarian).

With the diets of the first group, important theoretical results were obtained in the reduction of premature mortality (12% as mean value) and GHG emissions (up to 84%), and discrete results were found in cropland use and fertilizer use only in high-income countries, but negative global results were seen in freshwater use (increase of up to 16%) [76]. On the other hand, with vegetarian dietary patterns, in low-income countries, a decrease occurred only with regard to premature mortality and carbon footprint. In high- and medium-income countries, positive results were registered for all factors. For the latter, the drop in mortality was significant, between 19% (flexitarian diet) and 22% (vegan diet). Meanwhile, the reduction in environmental impact was progressively greater from the flexitarian diet to the vegan diet for all factors except for water footprint, which had an inverse trend. The reduction in values ranged between 54–87%, 2–11%, 8–11%, and 41–46%, respectively, for GHG emissions, freshwater use, cropland use, and fertilizer use (summing N footprint and P footprint) [76].

The strengths of this “global modelling analysis” are the simultaneous assessment of both environmental impacts and health outcomes (premature mortality due to chronic diseases) related to dietary patterns—adding significant evidence to the results reported in the previous paragraph—and the consideration of different socio-economic contexts (150 different countries were considered), thus broadening the view of the subject beyond national borders. It follows that: (a) both the sustainability and nutritional appropriateness of diets are context-related, and this additional level of complexity must always be kept in mind; (b) it would always be preferable to carry out investigations that assess health outcomes and environmental impacts together, as was done through the LCA by an important Italian study [77].

Further pertinent data, excluding the use of fertilizers, have been summarized in a recent review by Fresán and Sabaté [78], which is currently the most up-to-date source on this topic. Compared to the previous paper, the authors considered only lacto-ovo vegetarian (vegetarian) and vegan diets, and the analyses showed lower reductions in GHG emissions (35% and 49%, respectively), but greater decreases in (crop)land use (42% and 49.5%, respectively) and (fresh)water use (28%, only vegetarian). Another difference was represented by the increase—although not significant—in the water footprint associated with a vegan diet compared to that of an omnivorous one.

All of the above GHG emissions data show the same trend as the analyses of the two main vegetarian cohorts, AHS-2 and EPIC-Oxford [79,80]. Therefore, it has been largely demonstrated that vegetarian patterns have, to a different extent, a lesser negative impact than omnivorous diets. From this perspective, they could constitute a valid solution to the aforementioned trilemma [81], were there to be a population-level dietary change.

The last necessary consideration regards the high interindividual variability among vegetarian diets: in order not to make false considerations, one should not always rely on average nominal diets (diet scenarios). Rather, it is necessary to evaluate the foods actually consumed within individual vegetarian diets, since in some cases, it is possible that these patterns have a greater total environmental impact than those of some omnivores [82]. This is the reason why such dietary patterns are not the only solution for this huge public health problem. Other healthy dietary patterns (such as an effective Mediterranean diet and the DASH diet) are certainly further arrows in our bow to counteract the environmental impact overall.

### 4.2. Mediterranean Diet

#### 4.2.1. Health Outcomes

Table 2 summarizes the main positive health outcomes associated with the Mediterranean diet.

The Mediterranean diet has not only been considered as one of the main elements of a good lifestyle, and therefore as a primary prevention tool; several interventional studies have shown that this dietary pattern is an effective therapeutic option even in secondary prevention.

#### 4.2.2. Environmental Impact

Some studies have also been carried out on environmental sustainability, comparing the effects of Mediterranean and vegetarian diets. The previously cited review [78] stated that a healthy Mediterranean diet, as well as some national dietary guideline-related diets, contributed significantly less to a reduction in GHG emissions than vegetarian patterns (10% versus the 22–87% range seen in the previous paragraph), contrary to the US Scientific Report of 2015 DGAC [26] and its subsequent review [92] that placed the diets on the same level.

The role of the Mediterranean diet in the challenge against climate change is, however, noteworthy, given its reduced effect on several factors of the “environmental pressure” equation. As early as 2008, the Mediterranean diet was defined as «rich in biodiversity and nutritionally healthy» [93], and three years later, it was referred to as an example of a sustainable diet [94].

Regarding GHG emissions, in 2013, Coop Italia proposed the Environmental Hourglass Model [95,96]. This represents the weekly carbon footprint of a person who follows the nutritional indications of the Mediterranean diet model, more precisely, the 2003 CRA-NUT guidelines [97]. Although an Italian study from 2017 defined the model as a useful tool «to help translate health-promoting dietary recommendations that consider regional circumstances and cultural diversity into practical eating habits, to promote sustainable and environmentally friendly consumption» [98], it is also true that the risk of underestimating the environmental impact is real, since the model is based on the assumption that nutritional guidelines are followed, but unfortunately, this is often not the case [99].

To extend the discussion to the other environmental impact factors listed in the first paragraph (the various environmental footprints), it is useful to mention a Spanish cohort study [100]. More precisely, together with GHG emissions, (crop)land use, water use, and energy demand were also included. The objective of the study was to evaluate the effect of greater adherence to the Mediterranean diet (using the Mediterranean Diet Score-2 (MDS-2) [91]) on the “environmental pressure” factors considered both separately and as a whole.

The most graphically appealing example of the inextricable link between food sources and environmental sustainability is provided by the “double pyramid”, in which the Mediterranean diet pyramid is drawn up against an environmental impact pyramid (Figure 1). This scheme was proposed for the first time in 2010 by the Barilla Center for Food and Nutrition (BCFN) Foundation and was updated in 2021 [101]. The key message is that «in practice, two different but equally relevant goals—people’s health and environmental protection—fit into one single food model» [102].

Through these studies, the BCFN Foundation has shown that a menu based on meat has a total environmental impact that is two-and-a-half times greater than a vegetarian menu [103].

## 5. Health Outcomes and Environmental Impact of Protein Sources

The goal of this section is to investigate the different health outcomes and environmental impacts related to the various protein sources. In the nutritional field, it is important to deepen the different food macro-groups and their subcategories due to the great inter-individual variability of dietary patterns. In fact, as previously seen for vegetarian patterns [82], using hypothetical or predefined diets on the average population pattern (diet scenarios) is not always adequate and sometimes leads to evaluations that are incorrect. The use of the recent versions of the principal component analysis (PCA) method allows for the best evaluation of these dietary differences within the population considered in the studies [82].

Similarly, but in the opposite direction, the methodology used in the first decades of modern nutritional studies, which consists of an exclusive analysis of the effects of micro- and macronutrients considered separately from foods, is inadequate, because it is reductive and simplistic [104]. Foods should not be considered as a “vector of single beneficial micro- and macronutrients”, but as a complex mix of these in multiple combinations and with a fair variability, even between foods of the same category. Obviously, in the short term, evaluating any food in terms of all possible health outcomes and environmental impacts would be very difficult and a potential waste of time and resources. However, carrying out some analyses on principal food macro-groups, which straddle the line between nutrients and preset dietary patterns, can be a useful approach in the definition of increasingly valid and multidisciplinary nutritional guidelines [105].

Since the 1950s–1960s, there has been a phenomenon of “nutrition transition”, which, among other effects, has also caused a shift towards animal-based foods (pattern 4, “degenerative disease”) [106]. This has also led to the increased consumption of animal protein sources, favoring these over plant-based ones, an element that is strongly associated with economic and income growth [107,108].

The results of the systematic analysis of the Global Burden of Disease (GBD) Study 2017 on the health effects of dietary risks can be used to prove this statement. The study reported a higher-than-optimal level of consumption of animal protein sources (considering only meat and dairy products) for all the world’s high-income regions (including Western Europe), identified «as the level of risk exposure that minimizes the risk from all causes of death» [109]. In detail, the global intake of red meat is 18% higher than the optimal amount, but when considering only high-income regions, the value is, on average, double this amount. At the same time, in these regions, the consumption of plant-based protein sources (legumes, nuts, seeds, and whole grains) is, on average, below the optimal intake, with only some exceptions for nuts and seeds [109].

Nowadays, a huge amount of specific studies in the literature, but also many nutritional guidelines, emphasize the importance of a “nutrition transition” in the opposite direction (pattern 5, “behavioral change” [106]), referring more generally to plant-based foods, but also specifically to plant-based protein sources. Although this area of research is quite complicated, it has been widely shown that this dietary change would lead to positive effects on human health. Nonetheless, there would also be significant positive outcomes in terms of environmental sustainability, as demonstrated by the formerly reported study [75]. Once again, the importance of a multidisciplinary approach is evidenced, which, in this case, is guaranteed by a newborn field of studies—i.e., planetary health—that was proposed for the first time by the study conducted by the special “The Lancet—Rockefeller Foundation commission” [7].

Within this section, we have scrutinized various animal- and plant-based protein sources, highlighting positive and negative effects on human health by focusing on the most significant correlations and the most relevant chronic diseases. The following chronic diseases and groups of diseases were used for the analysis in terms of health outcomes: (a) incidence of CHD, stroke, HBP, chronic heart failure (CHF), T2D, overweight/obesity, MetS, colorectal cancer (CRC), breast cancer and gastric cancer; (b) all-cause mortality, total CVDs mortality, total CVDs incidence, total cancer mortality, and total cancer incidence.

Then, a more generic comparison was performed between animal and plant-based protein sources on their main health outcomes, reporting some risks of morbidity and mortality associated with a higher intake of some protein sources, or groups of them.

On the other hand, the environmental impact of specific protein sources is presented through the construction of a dedicated table at the end of this section, considering the six footprints already mentioned in this paper.

A fundamental premise concerns the selection of functional units to be used for the analysis of health outcomes and the environmental impact associated with the consumption of certain food groups [110]. If the use of functional nutritional units in the LCA at the level of individual foods is increasingly widespread for the area of food sustainability [111], in the health field, there is not the same homogeneity in the use of fixed quantities of a certain food (dietary serving sizes) for the evaluation of nutritional benefits in “dose-response” analyses.

Although it is certainly difficult to promote the use of the same approach globally, also given the great differences in terms of dietary patterns and eating habits worldwide, it would be appropriate to support its use to obtain reliable and evidence-based scientific data of ever-increasing quality.

In this analysis, an attempt has been made to compensate for this lack of homogeneity by taking as a reference the amount in grams that constitutes a serving and that is most widely used in the literature for each protein source evaluated. The various amounts are reported in each caption of the tables.

For the composition of the tables on the main health outcomes, priority was given to meta-analyses of prospective cohort studies or umbrella reviews of such meta-analyses; only in the absence of these types of papers, individual prospective cohort studies were considered, evaluating the quality and reliability of the data and always choosing the most recent sources whenever possible.

The results of some relevant RCTs or their systematic reviews or meta-analyses are reported in the text only to support the tables.

### 5.1. Animal Protein Sources

As seen earlier, protein quality can be evaluated in various ways. According to the traditional method linked to amino acid composition alone, and therefore to the ability of protein sources to provide protein synthesis while maintaining nitrogen balance, animal protein sources have higher values than plant-based ones with the DIAAS score currently used, and are therefore defined as “high-quality proteins”.

In this regard, the report of the EAT-Lancet Commission, which was based on the conclusions of a major study on cancer pathophysiology [112], states that an excessive contribution of high-quality proteins in adults could “maximally stimulate cell replication and growth”, thus increasing the risk of cancer [6].

Among the examined sources, there are two kinds of animal meat (fish, or more generally seafood, and animal meat) and two products of animal origin (eggs and dairy products).

#### 5.1.1. Meat

The first protein source analyzed is certainly the most well-studied in the literature. Regarding meat, there have been multiple studies in different fields of medicine that have highlighted the possible associations between some diseases and different levels of meat intake.

Meat can be classified using two different main criteria: the first is based on the origin of the product (red meat, poultry, and bushmeat); the second considers the processing (processed and unprocessed meat). Hence, there are a lot of different types of meat due to the various combinations deriving from these two classifications. Studying all of these possibilities is expensive from many points of view.

This study only considers red meat, processed meat, and total meat and poultry. As can be seen in Table 3, “total meat” often has less significant data compared to the other two groups chosen.

Red or processed meat exhibits many positive associations with chronic diseases, more so than any other food group analyzed. This can be explained by the higher content of SFAs in red meat [132]. These components have been associated with a series of molecular mechanisms that are involved in the main groups of chronic diseases, such as CVDs, metabolic diseases, and some cancers.

Meat-processing and some cooking methods play a role in increasing cancer risk. Processed meat, due to its higher concentrations of nitrates and nitrites, leads to a higher risk of GI tract cancers through the formation of N-nitroso-compounds [133]. Exposure to high temperatures during the cooking process is very harmful because of the formation of heterocyclic amines (HCAs) and polycyclic aromatic hydrocarbons (PAHs), which have been recognized as carcinogenic to humans and associated with increased CRC risk [133].

Regarding breast cancer, a significant positive association with meat intake (considering red meat and processed meat) was reported in the umbrella review by Farvid et al. [126]; the risk seems to be stronger in postmenopausal women.

In addition, underlining the increased risk of developing T2D that is associated with the consumption of an extra serving/day of meat, and especially the consumption of red meat and processed meat, a further increase in risk (41%) in the consumption of processed red meat was described, combining the results of eight previous meta-analyses [122]. Four of the eight meta-analyses considered [134,135,136,137] were the same as those mentioned in a previous review [138] that did not process the collected data further (and thus was not an umbrella review) but performed a comprehensive analysis of the complex pathophysiological mechanisms underlying the increased risk of T2D associated with a high/moderate intake of red and processed meat.

Among the various risk factors for T2D analyzed by another umbrella review [139], the consumption of processed meat (considered a dichotomous variable) was the worst among dietary risk factors, with a 41% increase in incidence rate.

#### 5.1.2. Fish

For the second category of animal protein sources, the term “seafood” has intentionally not been used, since marine foods of vegetable origin, such as algae and seaweeds, also fall within this terminology under a wider meaning.

Therefore, we have chosen a generic “fish” that is reflected in the “total fish” in Table 4; this group should be properly divided into “shellfish” and “finfish”, with analyses that separately evaluate these two food subgroups. If data are scant for the first group, the second can in turn be diversified into “oily fish (high-fat)” and “lean fish (low-fat)”, for which specific data have been reported. The reason for this further subdivision is to be found mainly in the difference in content of specific polyunsaturated fatty acids (PUFAs), such as “marine ω-3” ones, e.g., eicosapentaenoic acid (EPA) and docosahexaenoic acid (DHA). Some species of fish (precisely, every “oily fish”) have high levels of marine ω-3 acids, unlike “lean fish”.

Omega-3 fatty acids are one of the most crucial elements in the nutritional field, and there is much multidisciplinary interest around them [148,149]. The molecular mechanisms through which they act are multiple and varied; some are well-known, and others are still to be confirmed. Two not-so-recent reviews [150,151] present a clear picture of the action exerted within the human organism by these specific macronutrients: a reduction in inflammation (via the eicosanoid synthesis pathway) and LDL cholesterol concentrations are two of the main mechanisms demonstrated.

The main beneficial effects of omega-3 fatty acids concern cardiometabolic health, cancer, and brain health [152]. Positive outcomes related to a high or increased consumption of fish on cardiovascular health outcomes, except for HBP, are visible in Table 4.

Within the area of metabolic diseases, the beneficial effect of high plasma levels of omega-3 fatty acids on reducing the risk of overweight/obesity has been widely proven [153], but this does not automatically lead to similar improvements even with fish consumption. Further, in this specific case, there is a discrepancy between the level of macronutrients and that of food.

In fact, within Table 4, the data of a single meta-analysis have been inserted, and the results are not statistically significant. Indeed, they are nearly significant for an increase in the risk of overweight/obesity in a dose-response analysis [123]. Similarly, from the meta-analyses of prospective cohort studies, there is also no evidence on the risk of incidence of T2D and MetS associated with the consumption of this protein source [115,121,122,144], which is also due to the lack of such studies at a European level.

The only valid data in this field concern a reduction in the risk for abdominal obesity, one of the diagnostic elements of MetS: a significant decrease was found both in “highest vs. lowest intake” (relative risk (RR) 0.75, 95% confidence interval (C.I.) 0.62–0.89) and “dose-response” (RR 0.83, 95% C.I. 0.71–0.97) analyses [123].

Among the RCTs, the results are more promising: a meta-analysis has shown positive effects also related to oily fish intake (in addition to that of “marine omega-3”) both on the body weight and waist circumference [154].

However, it has not yet clearly been demonstrated that the increase in total fish consumption and its consumption instead of meat are associated with a reduction in the risk of obesity, MetS and T2D incidence, and the development of insulin resistance [155].

In addition, it is important to underline some data on the association with breast cancer. Although two recent meta-analyses of prospective cohort studies have not shown a positive association between total fish intake and such cancer [127,146], the second study and a previous review [146,152] reported the positive effect of “marine” omega-3 fatty acids (EPA and DHA) against this specific cancer. The reduction in incidence shown in the meta-analysis was significant in the “highest vs. lowest intake” analysis (RR 0.86, 95% C.I. 0.78–0.94) and nearly significant in the “dose-response” analysis (RR 0.95, 95% C.I. 0.90–1.00) [146].

Finally, a relevant paper from 2006 stated that the nutritional benefits of fish consumption are far greater than the related risks [156]. The important role of fish in cardiovascular health and neurodevelopment makes it an important food within the diet. The negative consequences related to the content of contaminants (such as mercury, other heavy metals, and dioxins and polychlorinated biphenyls) are currently considered less relevant than the health benefits of fish consumption.

In terms of fish protein, however, some issues are worth mentioning. One is that it is important to carry out appropriate research to monitor the marine water concentrations of the main heavy metals, particularly mercury [157] or plastics [158]. Indeed, the levels of mercury in the atmosphere and in oceanic waters have increased and must be monitored [159]. Further, even if the salubrious activities of omega-3 fatty acids have not been proven yet, fish consumption is advised [148,149]. This creates a large gap between supply and demand of these macronutrients due to the progressive depletion of wild fish [160]. A possible alternative is offered by nuts and seeds: their high content of alpha-linolenic acid (ALA, the only vegetable omega-3) might have positive effects on health [148,161,162,163].

#### 5.1.3. Eggs

The amino acid composition of the egg has been taken as a reference for the evaluation of protein quality with the most classic approach, given the great balance in the content of EAAs.

As for the association between egg intake and the main health outcomes, the trend is peculiar, and the data presented here in Table 5 give a clear idea about this trend. To best clarify this overview, “total CVDs incidence” was added to the evaluation of the outcome.

In brief, the literature reports: (a) a nearly significant increase in all-cause mortality [114,115]; (b) a non-significant reduction in total CVDs mortality [164] and stroke (indicated in two meta-analyses [164,165]), a significant reduction in total CVDs incidence [164], but at the same time, the absence of an association with CHD [115,118,131,164,165] and stroke (indicated in two other meta-analyses [115,118]), and an increased risk of CHF [118,164]. Data related to HBP indicate a strong significant reduction, but they are not very reliable given the low quality of evidence related to the fact that the large and important meta-analysis of reference [120] reported only one study for this outcome (unlike the others); (c) a not significant or conflicting result with both T2D [115,121,122,166] and CRC [115,124] risks.

The molecular mechanisms underlying these ambivalent outcomes are not yet fully understood and certainly need both further specific prospective studies and clinical trials; potential confounding variables such as sex, geographic area, and general dietary habits also need to be further investigated [164].

In addition, it is important to emphasize that eggs contain many macro- and micronutrients, as well as to debunk the belief that they are almost solely a source of cholesterol, and that their regular consumption is therefore associated with a high CV risk. A recent systematic review highlights that the CV effects of dietary cholesterol are not yet fully clarified and concludes by highlighting the importance of further «carefully adjusted and well-conducted cohort studies» [167].

#### 5.1.4. Dairy Products

As shown in Table 6, the consumption of total dairy products, but also of total milk and yogurt, is associated with a sharp reduction in MetS incidence both in “highest vs. lowest intake” and “dose-response” analyses [168]. Every single component of MetS evaluated in the study (five in total: hyperglycaemia, low HDL-cholesterol, hypertriacylglycerolaemia, abdominal obesity, and HBP) was reduced following the consumption of each of these three subgroups of dairy products in both types of analysis. Thus, this meta-analysis shows a significant decrease in abdominal obesity due to a higher consumption of total dairy products, total milk, and yogurt, respectively, of 24%, 17%, and 26%, making up for the low significance (for total dairy products) or the lack of data (for the other two subgroups) on overweight/obesity [168].

### 5.2. Plant-Based Protein Sources

Though plant-based protein sources sometimes contain fewer EAAs, people who consume a varied vegetarian diet can obtain the same quality of protein and similar amounts of nitrogen as those who consume animal protein sources or a mixed diet [172,173]. Among the sources we examined here are legumes, nuts and seeds, and cereal grains.

#### 5.2.1. Legumes

First of all, it is important to distinguish between the different types of legumes: pulses (dry legumes) and fresh legumes are the two main groups, and oilseed legumes (soybean and peanuts) have to be considered separately as peculiar types [174,175].

In Table 7, only “Total Legumes” and “Soybean” were analyzed due to the presence of more prospective cohort studies (and their meta-analyses) than the other groups named above.

However, there are many other studies in the literature demonstrating the numerous health benefits of pulses. Their high fiber content makes them low glycaemic index foods, suitable for the diet of diabetic subjects [181]. Furthermore, the improvement of the serum lipid profile associated with an adequate intake of this protein source leads to a reduction in CV risk due to several risk factors involved [181,182].

Although many leading health organizations recommend pulses as a fundamental element of the diet to reduce the risk of chronic diseases [181,183], the evidence is still insufficient, and all studies on the subject emphasize the urgency of higher-quality studies, both prospective cohort studies and randomized controlled trials [181,183,184,185].

#### 5.2.2. Nuts and Seeds

In this section, nuts and seeds are discussed together, because they are two food sub-groups with similar nutritional properties. Yet, though a good number of studies in line with the selection made for this review have been found for nuts, the same cannot be said for seeds: no prospective cohort studies (and, therefore, no meta-analyses) were retrieved from the literature. Instead, the following were found: (a) studies that considered nuts and seeds as a single class; among them, a prospective study on the cohort of AHS-2 showed how the high consumption of these foods (“highest vs. lowest intake” analysis) reduced the risk of CVDs mortality by 40% (C.I. 95% 0.42–0.86) [186]; (b) meta-analyses and systematic reviews of both observational and interventional studies that highlight a potential reduction in breast cancer risk associated with flaxseed intake, mainly due to lignans and the phenolic plant compounds contained in them in high amounts [187,188,189].

Even though Table 8 shows, in relation to other protein sources, data from meta-analyses from prospective cohort studies with incomplete results or with only nearly significant positive correlations in the association between nut consumption and the risk of major chronic diseases, this protein source has known benefits on some of the major health outcomes.

Nuts contain many healthful components, including unsaturated fatty acids (UFAs), dietary fiber, folate, antioxidants and vitamins (vitamin E and tocopherols), minerals (such as magnesium and potassium), and phytochemicals (e.g., flavonoids), which, «in isolation or as part of enriched foods, improve cardiometabolic risk factors» [177]. Many controlled trials showed positive effects on CVD risk factors [130], and the PREDIMED study showed the importance of nut consumption in reducing the risk of cardiovascular events [191]. A recent meta-analysis of five prospective cohort studies shows that an adequate consumption of nuts has been positively associated with both primary and secondary endpoints [190]. In addition to the significant decrease in the risk of overweight/obesity incidence (primary), nuts also reduced the risk of weight gain and increased waist circumference. The same study carried out a meta-analysis of many RCTs, showing that there is no weight gain associated with a consumption of nuts that is not excessive (below the maximum limit identified by the main nutritional guidelines). These data confirm some results that were previously published [192,193], as well as how unfounded the perception is that the intake of this protein source is related to weight gain. This is probably due to the high content of fatty acids in nuts, but actually, this element is one of their strengths: nuts are high-density foods, and this is associated with greater satiety. In addition, one of the main components of nuts is monounsaturated fatty acids (MUFAs)—and in some cases, PUFAs, especially in walnuts—which, through various mechanisms, have an important role in improving CV health [148]. Concerning this, the authors showed that the intake of MUFAs and PUFAs of plant origin (especially the only omega-3 PUFA, ALA) had clear advantages over the consumption of SFAs of animal origin [148].

A specific analysis on ALA was proposed, and the prospects of its role in the prevention of cardiovascular and metabolic diseases are interesting [148]. Future specific research will have to be focused on the conversion efficiency of ALA into EPA and DHA, the two omega-3 fatty acids that our body uses for various functions.

From a quantitative point of view, the content of ALA in nuts, flax seeds, and chia seeds is greater than that contained in all species of oily fish [194].

#### 5.2.3. Cereal Grains

Cereal grains, despite having the lowest mean protein content among the sources we selected, are considered among plant-based protein sources; they are the main source of proteins in some low- and middle-income countries, and their preferential use is associated with better prognosis. The latter is a further demonstration of the fact that the beneficial effects of protein sources are not only related to their protein content, but to all the nutrients contained in them, and that “nutritionism” [195] should be avoided.

One of the most noteworthy inverse associations in Table 9 is the reduction in T2D risk. Two recent umbrella reviews confirmed the results of 12 and 2 previous meta-analyses, respectively [122,139], and showed (the first one with high-quality evidence) a significant reduction in the risk of T2D (13%) associated with a serving-per-day increase in whole grains.

### 5.3. Protein Sources Comparison

The two macro-categories of protein sources can be compared in terms of health outcomes or environmental impact, keeping in mind that the two issues are now intertwined.

Data on the mortality of the main chronic diseases and the incidence risks associated with the consumption of one or the other macro-category of protein sources are less important than the data reported for individual protein sources; «generalizing the health benefits of plant protein over animal protein is difficult due to trial inconsistencies and limited control of variables» [197]. Therefore, the dichotomization of protein sources into animal- and plant-based sources leads to less valid results, which is not in contrast to the widely demonstrated benefits associated with the consumption of plant-based protein sources [198]. It simply suggests that studies and analyses must be carried out primarily by focusing research on individual protein sources and not on the whole macro-category.

For the assessment of environmental impact, however, this dichotomy is not associated with a lower impact, because there is a major difference between the two macro-categories in terms of their total environmental impact.

#### 5.3.1. Health Outcomes

The following discussion is divided into two parts. The first concerns the data on the various types of mortality; the second concerns the data on the incidence of the main chronic diseases examined (Table 10).

##### Mortality

Data on all-cause mortality and CVDs mortality related to the consumption of the two different macro-categories of protein sources—reported in Table 10 from the analysis of two of the largest US cohorts (“Nurses Health Study” and “Health Professionals Follow-up Study”) [199]—were split into two groups of participants. The data were statistically significant (showing an inverse association) for plant-based sources and nearly significant for animal ones only among those individuals with at least one unhealthy lifestyle factor (smoking, alcohol consumption, physical inactivity, and overweight/obesity were considered): (a) increased mortality due to a larger intake of animal protein sources, 1.03 (C.I. 95% 0.99–1.07) for all-cause and 1.08 (C.I. 95% 1.00–1.17) for CVDs; (b) reduction in mortality related to a higher consumption of plant-based proteins, 0.90 (C.I. 95% 0.85–0.95) for all-cause and 0.88 (C.I. 95% 0.79–0.98) for CVDs.

The same study confirmed, with evidence of higher quality and greater completeness, the results of a previous paper on the reduction in CVDs mortality associated with the substitution of various animal protein sources with plant-based ones [203]. Moreover, the hazard ratios (HRs) of other causes of death related to this protein source replacement for three percent total energy were highlighted [199].

In terms of cancer mortality, there appears to be no significant differences between the two macro-categories [199]. A subsequent systematic review [200] confirmed the data obtained from the previous study almost entirely, both those related to the three mortality groups for the two different macro-categories of protein sources and those for the substitution of animal protein sources with plant-based ones for three percent of the total energy ingested.

The substitution of animal protein sources with plant-based protein sources is another method by which the associations between the intake of dietary protein sources and health outcomes can be evaluated. The use of “highest vs. lowest intake” and “dose-response” analyses is not so effective when comparing two macro-groups that contain foods with different characteristics. There are few studies in the scientific literature that carry out such analyses, and for this reason, Table 10 and Table 11 present fewer data than those on individual protein sources.

Within the systematic analysis of the GBD Study from 2017 cited above [109], mortality and DALYs related to dietary risks have been extensively analyzed. Regarding Western Europe, it emerged that about 40–45% of both mortality and DALYs is related to imbalances in the intake of protein sources (considering whole grains as a plant-based protein source). One consideration to be made in light of the results of this analysis is that the negative health outcomes (considering mortality and DALYs) linked to too low a consumption of plant-based protein sources are much greater than those of an excessive intake of animal protein sources. This should be an important public health message to encourage the transition to an increased consumption of plant-based protein sources.

##### Incidence

A noteworthy report by Mozaffarian, Appel, and Van Horn [130] collected a series of prospective cohort studies and RCTs on clinical CV endpoints (CHD, stroke, or T2D) and the main CV risk factors associated with the intake of specific food groups. Considering only the evaluated protein sources, it can be seen that the consumption of nuts and whole grains, among the plant-based sources, and of fish and dairy products, among the animal sources, were associated with a reduction in CV risks and in the incidence of the abovementioned clinical conditions, with substantial evidence included. On the contrary, the consumption of unprocessed red meat and processed meat increased the risk of CVDs in almost all studies, other than the insubstantial effect of unprocessed red meat found that was found without much evidence in prospective cohort studies. Further studies have been carried out over the last decade, but more are needed to obtain clearer and more reliable data.

Within the analysis of each protein source, no specific data were reported on their associations with neurodegenerative diseases. Referring to dietary patterns, a major study from 2015 showed that there is a reduction in the incidence of Alzheimer’s associated with all three patterns considered in the paper. If for Mediterranean and DASH diets, the reduction is significant only when comparing “highest vs. lowest intake” adherence diet scores (in tertiles), for the Mediterranean-DASH Intervention for Neurodegenerative Delay (MIND) diet, it is also significant for the intermediate tertile (moderate adherence) [204].

This dietary pattern is characterized by a high consumption of fish as an important source of omega-3 fatty acids, as well as by a greater intake of plant-based foods and of «phytonutrient-rich foods, such as berries and green leafy vegetables, which have been demonstrated to have neuroprotective benefits» [205].

Given their neuroprotective effects [152], it is plausible to propose a role for fish as a suitable protein source for the prevention of neurodegenerative diseases, in addition to its protective effect against the main CVDs (CHD, stroke, and CHF of all) already highlighted in the dedicated section above.

Already in 2002, the final paper of a French cohort study showed a significant inverse association between dementia (classified at the time with the obsolete Diagnostic and Statistical Manual (DSM) of Mental Disorders (DSM-III-R)) and the consumption of at least one serving per week of fish; the identified risk reduction was 34% (HR 0.64, 95% C.I. 0.47–0.93) [206]. This data, although limited to a single cohort, as well as being outdated and not of high quality, was included in a more recent review that associated various nutritional elements to the risk of cognitive impairment [207]. Among protein sources, fish is the one with the largest number of observational studies that have demonstrated a protective effect (five out of seven studies made this observation).

Another protein source evaluated in the same review that may have a potential role in this clinical area was nuts, since they contain another omega-3 PUFA previously named, i.e., ALA. Although the studies reported here showing a protective effect only comprised two small trials, in which the consumption of olive oil was not evaluated separately, in the scientific literature, there are several papers that have shown the beneficial effects of this omega-3 fatty acid [148,161,162].

#### 5.3.2. Environmental Impact

As previously mentioned, it is also important to consider the environmental impact of protein sources from a planetary health perspective [208,209].

It is very important to underscore that planetary health is inseparable from human health. In other words, the best diet is “good for you, good for the planet” and can be visually represented by the double pyramid developed by the BCFN Foundation (Figure 1). In this respect, plant-based protein sources are much more sustainable than their animal-based counterparts [208,210]. An Italian study calculated that the adoption of a sustainable and healthy diet had a 47% lower carbon footprint and 25% lower water footprint than an unhealthy diet, while impacting income and food monthly expenditures, on average, 13% less [211]. Further, the desirable diet had a 21% lower impact on the sanitary costs related to CVDs [211].

In addition to the GHG emissions along the entire life cycle of a product, the environmental impact in the food sector also refers to the consumption of natural or artificial resources and the consequences on the ecosystem. Therefore, the amounts of water and soil consumed and that of the fertilizers and energy needed for production, which are associated with the use of potentially harmful substances (such as pesticides and drugs) and the loss of biodiversity as an index of damage to the natural environment, can return a fairly clear idea of the “environmental cost” of a food or group of foods. For a more detailed description of the various types of footprints, see the paper previously cited [23].

As for the health field strictly, it is also possible in this case to carry out quantitative analyses through multiple units of measurement. Thus, the data can also slightly change depending on the functional reference. Taking the example of carbon footprint, in Table 11, we only inserted data in kg CO_2_-eq/kg food (or food group). In the literature, though, there are studies that refer to the standard units of food (g or kg CO_2_-eq/serving), the caloric intake (g or kg CO_2_-eq/kcal), or only to the amount of protein (g or kg CO_2_-eq/g protein) [70]. Such a lack of uniformity certainly complicates the assessment of these various environmental impacts. Relying on the consideration that protein sources are not only made up of proteins, and in order to maintain a correlation with the analyses presented in the previous section, the first two units of measurement were chosen for reference, thus referring to the food weight (net or in preselected servings).

**Table 11 nutrients-14-05115-t011:** Environmental footprints of various protein sources.

Environmental Footprints		Units of Measurement	Red Meat	Poultry	FISH	EGGS
Carbon footprint		kg CO_2_-eq/kg (food)	25.58/26.61 [212] ^a^ 5.77 [212] ^b^	3.65 [212]	3.49 [212]	3.46 [212]
Water footprint (total)		m^3^/ton (food)	8761/15415 [213] ^a^ 5988 [213] ^b^	4325 [213]	1974 [214]	3265 [213]
Land footprint		m^2^/kg (food)	308.58/542.82 [215] ^a^ 19.53 [215] ^b^	19.22 [215]	0–10 [216]	17.83 [215]
CED ^e^		MJ/kg (food)	37–82 [217] ^a^ 25–31 [217] ^b^	18–33 [217]	No data	12–17 [217]
Use of chemicals	Fertilizers (N footprint and P footprint)	10 g N/serving	30.01/30.27 [6] ^a^ 56.68 [6] ^b^	55.22 [6]	18.46 [6]	25.61 [6]
		10 g P/serving	5.43/5.89 [6] ^a^ 9.75 [6] ^b^	9.92 [6]	3.98 [6]	4.40 [6]
	Pesticides	/	No data	No data	No data	No data
Biodiversity footprint		/ ^f^	VERY HIGH [218]	VERY HIGH [218]	VERY HIGH [110]	HIGH [110]
**Environmental** **footprints**		**Units of measurement**	**Dairy** **products**	**Legumes**	**Nuts**	**Cereal grains**
Carbon footprint		kg CO_2_-eq/kg (food)	8.55/9.25 [212] ^c.^ 1.29 [212] ^d^/2.59 [219] ^d^	1.20 [212]	0.51 [212]	0.50 [212]
Water footprint (total)		m^3^/ton (food)	5553/6760 [213] ^c^ 1020 [213] ^d^/1485 [219] ^d^	9063 [213]	4055 [213]	1644 [213]
Land footprint		m^2^/kg (food)	60.27/65.20 [215] ^c^ 9.09 [215] ^d^/12 [219] ^d^	6.96 [215]	11.19 [215]	2.81 [215]
CED ^e^		MJ/kg (food)	38 [217] ^c^ 3.0–3.1 [217] ^d^	2.9–7.4 [217]	No data	1.7–9.6 [217]
Use of chemicals	Fertilizers (N footprint and P footprint)	10 g N/serving	15.18 [6]	0 [6]	4.28 [6]	No data
		10 g P/serving	3.79 [6]	0 [6]	0.63 [6]	No data
	Pesticides	/	No data	No data	No data	No data
Biodiversity footprint		/ ^f^	high [110]	low [110]	high [110]	intermediate [220] ^g^

^a^ Refers to beef; ^b^ refers to pork; ^c^ refers to cheese/butter (dairy products with higher fat content); ^d^ refers to milk/yogurt (dairy products with lower fat content); ^e^ ranges of data have been included for this footprint, because in the study from which they were extrapolated (a study that is among the most recent and authoritative), only specific foods were considered and not the food groups mentioned in this table; ^f^ VERY HIGH = strong negative impact; HIGH = moderate negative impact; INTERMEDIATE = neutral impact; LOW = positive impact. Scale made taking as reference specific table of the HCWH report “Redefining protein: adjusting diets to impact public health and conserve resources” [110]; ^g^ whole grains must be differentiated from refined ones, as the impact of the former is decidedly less.

In any case, Table 11 clearly shows a more dichotomous trend between animal- and plant-based protein sources than the previous analysis of health outcomes.

Protein efficiency is much higher for foods of plant-based origin than those of animal origin [217]. The two groups of protein sources with the greatest overall environmental impact are meat and dairy products, mainly because of livestock production [221,222].

To give an example of how much greater the “environmental pressure” of animal protein sources is compared to plant-based ones, a quote is reported here comparing the different environmental footprints associated with kidney beans and various animal sources: «to produce 1 kg of protein from kidney beans required approximately eighteen times less land, ten times less water, nine times less fuel, twelve times less fertilizers and ten times less pesticide in comparison to producing 1 kg of protein from beef. Compared with producing 1 kg of protein from chicken and eggs, beef generated five to six times more waste (manure) to produce 1 kg of protein.» [223].

At the same time, it is important to note that the water footprint values of the various protein sources shown in Table 11 are in line with that which was previously stated in the analysis of the environmental impact of vegetarian dietary patterns [76]. Among the environmental footprints considered, the smallest positive differences between the omnivorous diet (following the nutritional claims) and the various vegetarian patterns were reported in the water footprint. The plant-based diets included in the “diet scenarios” of the same study had even higher water use than the omnivorous diet taken as a reference [76]. The data in Table 11 are a further confirmation of this trend.

An element that stands out is that legumes have zero impact on the footprints related to the use of fertilizers. Moreover, this is the only protein source that has a positive biodiversity footprint, since it leads to an increase in so-called agrobiodiversity. These two features are linked to their ability to fix nitrogen and stimulate microbial activity in soil, respectively [175].

Although it is a factor of fundamental importance, we found no studies on pesticides to be included in Table 11 [110]. Another element relevant to the planetary health model that could be added into the “use of chemicals” factor is the use of antibiotics associated with the production of these protein sources. One of the main public health issues in the coming decades will be antimicrobial resistance (AMR) [224].

Finally, it is necessary to stress how these different environmental impacts also, whether directly or indirectly, cause multiple health consequences with mechanisms and processes that are not explored here, given the high complexity of the subject.

## 6. Conclusions

Several prospective cohort studies, some meta-analyses, and an umbrella review of various meta-analyses have shown that the use of preferential vegetable protein sources is associated with a better prognosis in terms of major metabolic diseases and CVDs as compared with the intake of animal protein sources. At the same time, no differences were demonstrated between the two types of protein sources in terms of muscle and bone health, and there are some clinical conditions in which a vegetarian diet might increase the risk of, e.g., vitamin B12 deficiency and the related reduced functioning of one-carbon metabolism, zinc deficiency, and hypoferritinemia.

With a view to planetary health, it is necessary to consider the overall “environmental pressure” of food production also in nutrition claims; for this reason, a synthesis of the main environmental impact factors of the various protein sources was carried out. It can be noted that animal protein sources generally have a greater environmental impact than plant-based ones, and therefore, a comparison between the two macro-categories is more appropriate than strictly the nutritional field. Though several multidisciplinary studies have extensively analyzed the issue of sustainable nutrition, there are a lack of tools in the medical health field that allow us to apply this vision of global health also to a nutritional prescription for patients.

In conclusion, the data discussed in this paper allow us to conclude that the consumption of vegetable protein sources is associated with better health outcomes (namely, on the cardiovascular system) than animal-based product use. As far as mechanisms of action are concerned, there are currently no data to explain these effects, and much more research is needed, e.g., on digestibility [9]. However, the irrefutably healthier outcomes of vegetable protein sources dovetail with their lower environmental impact, which must be considered when designing optimal diets. The health of the planet cannot be disjointed from the health of the human being.

Future research will clarify the putative health effects of vegetable protein sources when compared with animal ones, and it should foster better agronomic practices and influence public health in a direction that will benefit both the planet and its inhabitants.

## Figures and Tables

**Figure 1 nutrients-14-05115-f001:**
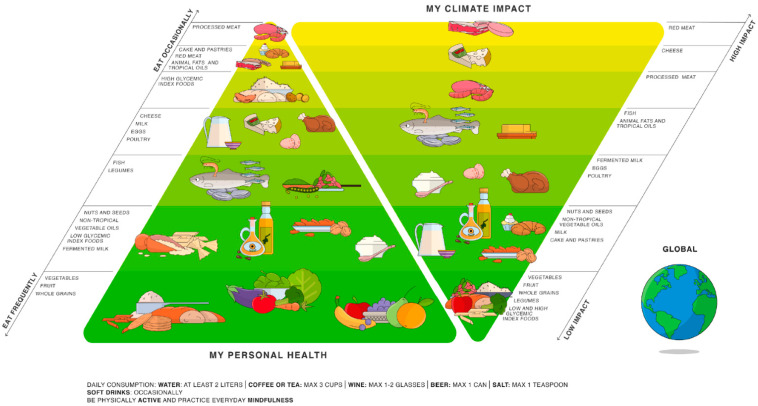
The Global Double Pyramid. As one health approach to food, the Double Pyramid connects food culture, health, and climate. Source: Barilla Foundation and Research Unit on Nutrition, Diabetes, and Metabolism, University of Naples Federico II, 2021.

**Table 1 nutrients-14-05115-t001:** All-cause mortality of vegetarian dietary patterns compared to the omnivorous reference diet.

References	Year	Vegetarian Dietary Patterns				
		Vegetarian (Total)	Flexitarian	Pescatarian	Lacto-Ovo Vegetarian	Vegan
[64]	1999	0.95 (0.82–1.11)	0.84 (0.77–0.90)	0.82 (0.77–0.96)	0.84 (0.74–0.96)	1.00 (0.70–1.44)
[57]	2009	1.05 (0.93–1.19)	No data	0.89 (0.75–1.05)	1.03 (0.90–1.16)	
[42]	2012	0.91 (0.66–1.16)	No data	No data	No data	No data
[32,50]	2013/2014	0.88 (0.80–0.97)	0.92 (0.75–1.13)	0.81 (0.69–0.94)	0.91 (0.82–1.00)	0.85 (0.73–1.01)

Legend of backgrounds: white = negative/positive statistically significant association; light gray = negative/positive statistically nearly significant association; gray = statistically non-significant association/absence of association; dark gray = missing value.

**Table 2 nutrients-14-05115-t002:** The most relevant health benefits from good adherence to the Mediterranean diet found in some of the main prospective studies of the last decade. Data inserted refer to a fixed-effects model.

Health Outcomes	Unit of Intake	RR (95% C.I.)	N° of Prospective Studies	References
Mediterranean Diet				
All-cause mortality	Highest vs. Lowest	No data		
	2-Point Increase MDS	0.92 (0.91–0.93)	7	[83]
		0.90 (0.89–0.91)	28	[84]
Total CVDs mortality	Highest vs. Lowest	0.79 (0.77–0.82)	21	[85]
	2-Point Increase MDS	0.91 (0.87–0.96)	26	
Total CVDs incidence	Highest vs. Lowest	/	8	
	2-Point Increase MDS	0.90 (0.87–0.92) ^a^	8	[83]
		0.90 (0.85–0.96)	12	[85]
CHD mortality	Highest vs. Lowest	0.73 (0.59–0.89)	6	
	2-Point Increase MDS	0.94 (0.91–0.96)	6	
CHD incidence	Highest vs. Lowest	0.73 (0.62–0.86)	7	
	2-Point Increase MDS	0.80 (0.76–0.85)	8	
Stroke mortality	Highest vs. Lowest	0.87 (0.80–0.96)	4	
	2-Point Increase MDS	/	6	
Stroke incidence	Highest vs. Lowest	0.80 (0.71–0.90)	5	
	2-Point Increase MDS	0.90 (0.85–0.96)	10	
HBP	Highest vs. Lowest	No data		
	2-Point Increase MDS	No data		
CHF	Highest vs. Lowest	No data		
	2-Point Increase MDS	No data		
T2D	Highest vs. Lowest	0.87 (0.82–0.93)	6	[86]
	2-Point Increase MDS	No data		
Overweight/Obesity	Highest vs. Lowest	No data		
	2-Point Increase MDS	No data		
MetS	Highest vs. Lowest	0.73 (0.54–0.98)	4	[87]
	2-Point Increase MDS	No data		
NDDs	Highest vs. Lowest	0.74 (0.65–0.84) ^b^	12	[88]
	2-Point Increase MDS	0.87 (0.81–0.94)	4	[89]
Total cancer mortality	Highest vs. Lowest	0.87 (0.82–0.92)	18	[90]
	2-Point Increase MDS	No data		
Total cancer incidence	Highest vs. Lowest	No data		
	2-Point Increase MDS	0.96 (0.95–0.97) ^a^	8	[83]
CRC ^c^	Highest vs. Lowest	0.92 (0.87–0.99)	10	[90]
	2-Point Increase MDS	No data		
Breast cancer ^c^	Highest vs. Lowest	0.97 (0.94–1.00)	12	[90]
	2-Point Increase MDS	No data		
Gastric cancer ^c^	Highest vs. Lowest	0.77 (0.64–0.92) ^d^	4	[90]
	2-Point Increase MDS	No data		
Respiratory tract cancers ^c^	Highest vs. Lowest	0.84 (0.76–0.94) ^d^	5	[90]
	2-Point Increase MDS	No data		

^a^: Combination of mortality and incidence; ^b^: in this case, the risk of cognitive impairment, any type of dementia, and Alzheimer’s dementia, specifically, were considered at the same time; ^c^: for most of the site-specific cancers, as well as for inflammatory and metabolic parameters, the evidence was interpreted as only suggestive and weak, and further research is needed for these outcomes [88]; ^d^: compared to breast and colorectal cancers, the number of observational studies considered is lower. CHD = coronary heart disease; CHF = chronic heart failure; CRC = colorectal cancer; CVDs = cardiovascular diseases; HBP = high blood pressure; MDS = Mediterranean diet score [91]; MetS = metabolic syndrome; NDDs = neurodegenerative diseases/disorders; T2D = type 2 diabetes.

**Table 3 nutrients-14-05115-t003:** Association between red meat, processed meat, and total meat and poultry intake and incidence of some major chronic diseases/mortality of major disease groups (RR, 95% C.I.). The dose-response meta-analysis always refers to an additional serving of a food per day relative to the average intake of that food observed in a cohort study (100 g for total meat, red meat, and poultry; 50 g for processed meat).

Health Outcomes	Unit of Intake	RR (95% C.I.)	N° of Prospective Studies	References
**Red Meat**				
All-cause mortality	Highest vs. Lowest	/	2	[113]
		1.10 (1.00–1.22)	12	[114]
	Dose-Response	1.10 (1.04–1.18)	10	
		1.12 (1.05–1.21)	/	[115]
Total CVDs mortality	Highest vs. Lowest	0.88 (0.77–1.01) ^a^	8	[116]
		1.16 (1.03–1.32)	5	[117]
	Dose-Response	1.15 (1.05–1.26)	3	
CHD	Highest vs. Lowest	1.15 (1.08–1.23)	/	[115]
		1.16 (1.08–1.24)	3	[118]
	Dose-Response	1.15 (1.08–1.23)	3	
Stroke	Highest vs. Lowest	1.16 (1.08–1.25)	7	
		1.11 (1.03–1.20)	3	[119]
	Dose-Response	1.12 (1.06–1.17)	7 /	[118] [115]
HBP	Highest vs. Lowest	1.15 (1.02–1.28)	7	[120]
	Dose-Response	1.14 (1.02–1.28)	7	
CHF	Highest vs. Lowest	1.12 (1.04–1.21)	5	[118]
	Dose-Response	1.08 (1.02–1.14)	4	
T2D	Highest vs. Lowest	1.22 (1.10–1.36)	13	[121]
	Dose-Response	1.17 (1.08–1.26)	14	[122]
		1.13 (1.03–1.23)	/	[115]
Overweight/obesity	Highest vs. Lowest	1.23 (1.07–1.41)	1	[123]
	Dose-Response	No data		
MetS	Highest vs. Lowest	No data		
	Dose-Response	No data		
Total cancer mortality	Highest vs. Lowest	/	2	[113]
	Dose-Response	No data		
Total cancer incidence	Highest vs. Lowest	No data		
	Dose-Response	No data		
CRC	Highest vs. Lowest	1.12 (1.06–1.18)	25	[124]
	Dose-Response	1.12 (1.06–1.19)	18 /	[124] [115]
		1.12 (1.00–1.25)	8	[125]
BREAST CANCER	Highest vs. Lowest	1.09 (0.99–1.21)	7	[126]
		/	8	[127]
	Dose-Response	1.07 (1.01–1.14)	6	
GASTRIC CANCER	Highest vs. Lowest	/ /	13 6	[128] [129]
	Dose-Response	/	4	[129]
**Processed Meat**				
All-cause mortality	Highest vs. Lowest	1.21 (1.16–1.26)	7	[114]
	Dose-Response	1.23 (1.12–1.36)	7	
		1.41 (1.21–1.67)	/	[115]
Total CVD mortality	Highest vs. Lowest	0.81 (0.75–0.87)	/	[116]
		1.18 (1.05–1.32)	7	[117]
	Dose-Response	1.24 (1.09–1.40)	6	
		1.15 (1.07–1.24)	6	[113]
CHD	Highest vs. Lowest	1.15 (0.99–1.33)	5	[118]
	Dose-Response	1.27 (1.09–1.49)	3 /	[118] [115]
		1.42 (1.07–1.89)	6	[130]
STROKE	Highest vs. Lowest	1.16 (1.07–1.26)	6	[118]
		1.17 (1.08–1.25)	4	[119]
	Dose-Response	1.17 (1.02–1.34)	6 /	[118] [115]
HBP	Highest vs. Lowest	1.12 (1.02–1.23)	5	[120]
	Dose-Response	1.12 (1.00–1.26)	4	
CHF	Highest vs. Lowest	1.27 (1.14–1.41)	3	[118]
	Dose-Response	1.12 (1.05–1.19)	2	
T2D	Highest vs. Lowest	1.39 (1.29–1.49)	11	[121]
	Dose-Response	1.32 (1.19–1.48)	/	[115]
		1.37 (1.22–1.54)	14	[122]
		1.57 (1.28–1.93)	8	[130]
Overweight/Obesity	Highest vs. Lowest	No data		
	Dose-Response	No data		
MetS	Highest vs. Lowest	No data		
	Dose-Response	No data		
Total cancer mortality	Highest vs. Lowest	No data		
	Dose-Response	1.08 (1.06–1.11)	5	[113]
Total cancer incidence	Highest vs. Lowest	No data		
	Dose-Response	No data		
CRC	Highest vs. Lowest	1.14 (1.06–1.21)	21	[124]
	Dose-Response	1.17 (1.10–1.23)	16 /	[124] [115]
		1.18 (1.10–1.28)	10	[125]
Breast cancer	Highest vs. Lowest	1.09 (1.03–1.16)	15	[126]
		1.07 (1.01–1.14)	14	[127]
	Dose-Response	1.09 (1.02–1.17)	12	
Gastric cancer	Highest vs. Lowest	1.15 (1.03–1.29)	8	[128]
		1.24 (1.04–1.47)	10	[129]
	Dose-Response	1.21 (1.04–1.41)	7	
**Total Meat**				
All-cause mortality	Highest vs. Lowest	No data		
	Dose-Response	No data		
Total CVDs mortality	Highest vs. Lowest	/	6	[117]
	Dose-Response	/	6	
CHD	Highest vs. Lowest	1.23 (0.98–1.49)	7	[131]
	Dose-Response	No data		
Stroke	Highest vs. Lowest	1.18 (1.09–1.28)	4	[119]
	Dose-Response	No data		
HBP	Highest vs. Lowest	No data		
	Dose-Response	No data		
CHF	Highest vs. Lowest	No data		
	Dose-Response	No data		
T2D	Highest vs. Lowest	No data		
	Dose-Response	1.12 (1.01–1.24)	8	[122]
Overweight/Obesity	Highest vs. Lowest	No data		
	Dose-Response	No data		
MetS	Highest vs. Lowest	No data		
	Dose-Response	No data		
Total cancer mortality	Highest vs. Lowest	No data		
	Dose-Response	No data		
Total cancer incidence	Highest vs. Lowest	/	/	[90]
	Dose-Response	No data		
CRC	Highest vs. Lowest	No data		
	Dose-Response	No data		
Breast cancer	Highest vs. Lowest	No data		
	Dose-Response	No data		
Gastric cancer	Highest vs. Lowest	/	13	[128]
	Dose-Response	No data		
**Poultry**				
All-cause mortality	Highest vs. Lowest	No data		
	Dose-Response	/ ^b^	/	[115]
Total CVDs mortality	Highest vs. Lowest	/	6	[117]
	Dose-Response	/	5	
CHD	Highest vs. Lowest	No data		
	Dose-Response	/ ^b^	/	[115]
Stroke	Highest vs. Lowest	0.87 (0.78–0.96)	2	[119]
	Dose-Response	No data		
HBP	Highest vs. Lowest	No data		
	Dose-Response	No data		
CHF	Highest vs. Lowest	No data		
	Dose-Response	No data		
T2D	Highest vs. Lowest	No data		
	Dose-Response	/ ^b^ /	/ 3	[115] [122]
Overweight/Obesity	Highest vs. Lowest	No data		
	Dose-Response	No data		
MetS	Highest vs. Lowest	No data		
	Dose-Response	No data		
Total cancer mortality	Highest vs. Lowest	No data		
	Dose-Response	No data		
Total cancer incidence	Highest vs. Lowest	No data		
	Dose-Response	No data		
CRC	Highest vs. Lowest	No data		
	Dose-Response	0.78 (0.62–0.94) ^b^	/	[115]
		/	6	[125]
Breast cancer	Highest vs. Lowest	/	11	[127]
	Dose-Response	/	10	
Gastric cancer	Highest vs. Lowest	/ /	7 5	[128] [129]
	Dose-Response	/	4	[129]

^a^: Data obtained on lowest consumption, taking the highest amount as a reference, and therefore in contrast to all the others (“lowest vs. highest”); ^b^: data related to chicken consumption only and not to poultry.

**Table 4 nutrients-14-05115-t004:** Association between total fish, oily fish, and lean fish intake and some incidences of major chronic diseases/mortality of major disease groups (RR, 95% C.I.). The dose-response meta-analysis always refers to an additional serving of a food per day relative to the average intake of that food observed in a cohort study (100 g for total fish, oily fish, and lean fish).

Health Outcomes	Unit of Intake	RR (95% C.I.)	N° of Prospective Studies	References
**Total Fish**				
All-cause mortality	Highest vs. Lowest	0.95 (0.92–0.98)	38	[114]
		0.94 (0.90–0.98)	12	[140]
	Dose-Response	0.93 (0.88–0.98)	19 /	[114] [115]
		0.88 (0.83–0.93)	5	[140]
Total CVDs mortality	Highest vs. Lowest	No data		
	Dose-Response	No data		
CHD	Highest vs. Lowest	0.94 (0.88–1.02)	22	[118]
		0.81 (0.70–0.92)	29	[131]
		0.91 (0.84–0.97)	22	[141]
	Dose-Response	0.88 (0.79–0.99)	15 /	[118] [115]
Stroke	Highest vs. Lowest	0.95 (0.89–1.01)	20	[118]
		0.90 (0.85–0.96)	31	[142]
	Dose-Response	0.86 (0.75–0.99)	15 /	[118] [115]
		0.94 (0.89–0.99) ^a^	11	[143]
HBP	Highest vs. Lowest	/	8	[120]
	Dose-Response	1.07 (0.98–1.16)	7	
CHF	Highest vs. Lowest	0.89 (0.80–0.99)	8	[118]
	Dose-Response	0.80 (0.67–0.95)	7	
T2D	Highest vs. Lowest	/ /	9 7	[121] [122]
	Dose-Response	/	/	[115]
Overweight/Obesity	Highest vs. Lowest	/	1	[123]
	Dose-Response	1.06 (0.99–1.14)	1	
MetS	Highest vs. Lowest	No data		
	Dose-Response	0.80 (0.66–0.96) ^b^	6	[144]
Total cancer mortality	Highest vs. Lowest	No data		
	Dose-Response	No data		
Total cancer incidence	Highest vs. Lowest	0.98 (0.96–1.00)	/	[90]
	Dose-Response	No data		
CRC	Highest vs. Lowest	0.96 (0.90–1.01)	21	[124]
		0.93 (0.86–1.01)	22	[145]
	Dose-Response	0.93 (0.85–1.01)	16 /	[124] [115]
		0.89 (0.80–0.99)	11	[125]
Breast cancer	Highest vs. Lowest	/ /	11 18	[146] [127]
	Dose-Response	/	13	[127]
Gastric cancer	Highest vs. Lowest	/	10	[128]
	Dose-Response	No data		
**Oily Fish (Fat)**				
Stroke	Highest vs. Lowest	/	5	[147]
	Dose-Response	No data		
T2D	Highest vs. Lowest	0.89 (0.82–0.96)	4	[122]
	Dose-Response	No data		
**Lean Fish**				
Stroke	Highest vs. Lowest	0.81 (0.67–0.99)	4	[147]
	Dose-Response	No data		
T2D	Highest vs. Lowest	/	4	[122]
	Dose-Response	No data		

^a^: Per 3 servings/week vs. no intake; ^b^: any prospective cohort studies held in Europe.

**Table 5 nutrients-14-05115-t005:** Association between egg intake and incidences of some major chronic diseases/mortality of major disease groups (RR, 95% C.I.). The dose-response meta-analysis always refers to an additional serving of a food per day relative to the average intake of that food observed in a cohort study (50 g for eggs).

Health Outcomes	Unit of Intake	RR (95% C.I.)	N° of Prospective Studies	References
**Total Eggs**				
All-cause mortality	Highest vs. Lowest	1.06 (1.00–1.12)	8	[114]
	Dose-Response	1.15 (0.99–1.34)	5 /	[114] [115]
Total CVDs mortality	Highest vs. Lowest	No data		
	Dose-Response	0.95 (0.88–1.03)	8	[164]
Total CVDs incidence	Highest vs. Lowest	No data		
	Dose-Response	0.94 (0.89–0.99)	9	[164]
CHD	Highest vs. Lowest	/ /	6 11	[131] [118]
	Dose-Response	/ / / /	6 9 / 12	[165] [118] [115] [164]
Stroke	Highest vs. Lowest	/	10	[118]
	Dose-Response	/	10 /	[118] [115]
		0.97 (0.93–1.02)	6	[164]
		0.91 (0.81–1.02)	6	[165]
HBP	Highest vs. Lowest	0.54 (0.32–0.91)	1	[120]
	Dose-Response	0.25 (0.08–0.74)	1	
CHF	Highest vs. Lowest	1.25 (1.12–1.39)	4	[118]
	Dose-Response	1.16 (1.03–1.31)	4	
		1.11 (0.99–1.25)	4	[164]
T2D	Highest vs. Lowest	/	5	[121]
	Dose-Response	/ /	/ 13	[115] [122]
		1.16 (1.09–1.23)	13	[166]
Overweight/Obesity	Highest vs. Lowest	No data		
	Dose-Response	No data		
MetS	Highest vs. Lowest	No data		
	Dose-Response	No data		
Total cancer mortality	Highest vs. Lowest	No data		
	Dose-Response	No data		
Total cancer incidence	Highest vs. Lowest	No data		
	Dose-Response	No data		
CRC	Highest vs. Lowest	1.35 (1.11–1.36)	4	[124]
	Dose-Response	/ /	3 /	[124] [115]
Breast cancer	Highest vs. Lowest	/	9	[127]
	Dose-Response	/	8	
Gastric cancer	Highest vs. Lowest	/	9	[128]
	Dose-Response	No data		

**Table 6 nutrients-14-05115-t006:** Association between total dairy products, total milk, whole or skim milk, and yogurt intake and incidences of some major chronic diseases/mortality of major disease groups (RR, 95% C.I.). The dose-response meta-analysis always refers to an additional serving of a food per day relative to the average intake of that food observed in a cohort study (200 g for total dairy products and yogurt, 200 mL for total milk, whole milk, and skim milk).

Health Outcomes	Unit of Intake	RR (95% C.I.)	N° of Prospective Studies	References
**Total Dairy Products**				
All-cause mortality	Highest vs. Lowest	1.03 (0.98–1.07)	27	[114]
		/	33	[169]
	Dose-Response	0.98 (0.93–1.03)	16 /	[114] [115]
		0.99 (0.97–1.01)	20	[169]
Total CVDs mortality	Highest vs. Lowest	0.93 (0.88–0.98)	16	[169]
	Dose-Response	0.98 (0.96–1.00)	13	
CHD	Highest vs. Lowest	0.91 (0.82–1.00)	11	[130]
		/	13	[118]
	Dose-Response	/	10 /	[118] [115]
Stroke	Highest vs. Lowest	0.79 (0.75–0.82)	7	[130]
		0.96 (0.90–1.01)	12	[118]
	Dose-Response	0.98 (0.96–1.00)	11 /	[118] [115]
HBP	Highest vs. Lowest	0.89 (0.86–0.93)	9	[120]
	Dose-Response	0.95 (0.94–0.97)	9	
CHF	Highest vs. Lowest	/	3	[118]
	Dose-Response	1.08 (1.01–1.15)	1	
T2D	Highest vs. Lowest	0.89 (0.84–0.89)	11	[121]
		0.92 (0.86–0.97)	4	[130]
	Dose-Response	0.96 (0.94–0.99)	/ 21	[115] [122]
Overweight/Obesity	Highest vs. Lowest	/	6	[123]
	Dose-Response	0.97 (0.93–1.01)	5	
MetS	Highest vs. Lowest	0.75 (0.66–0.84)	12	[168]
	Dose-Response	0.91 (0.85–0.96)	9	
Total cancer mortality	Highest vs. Lowest	1.03 (0.98–1.07)	19	[169]
	Dose-Response	/	9	
Total cancer incidence	Highest vs. Lowest	0.95 (0.90–1.00)	/	[90]
	Dose-Response	No data		
CRC	Highest vs. Lowest	0.83 (0.76–0.89)	18	[124]
	Dose-Response	0.93 (0.91–0.94)	15 /	[124] [115]
		0.87 (0.83–0.90)	10	[125]
Breast cancer	Highest vs. Lowest	0.90 (0.83–0.98)	16	[170]
	Dose-Response	0.97 (0.95–0.99) ^a^	/	
Gastric cancer	Highest vs. Lowest	/	3	[128]
	Dose-Response	No data		
**Total Milk**				
All-cause Mortality	Highest vs. Lowest	/	27	[169]
	Dose-Response	1.03 (0.99–1.06)	16	
Total CVDs mortality	Highest vs. Lowest	/	15	[169]
	Dose-Response	/	9	
CHD	Highest vs. Lowest	No data		
	Dose-Response	No data		
Stroke	Highest vs. Lowest	No data		
	Dose-Response	No data		
HBP	Highest vs. Lowest	No data		
	Dose-Response	No data		
CHF	Highest vs. Lowest	No data		
	Dose-Response	No data		
T2D	Highest vs. Lowest	No data		
	Dose-Response	/	10	[122]
Overweight/Obesity	Highest vs. Lowest	No data		
	Dose-Response	No data		
MetS	Highest vs. Lowest	0.78 (0.69–0.87)	7	[168]
	Dose-Response	0.87 (0.79–0.95)	6	
Total cancer mortality	Highest vs. Lowest	/	13	[169]
	Dose-Response	1.03 (0.99–1.06)	8	
Total cancer incidence	Highest vs. Lowest	No data		
	Dose-Response	No data		
CRC	Highest vs. Lowest	No data		
	Dose-Response	0.94 (0.92–0.96)	9	[125]
Breast cancer	Highest vs. Lowest	0.94 (0.86–1.03)	/	[170]
		0.92 (0.84–1.02)	18	[127]
	Dose-Response	0.97 (0.93–1.01)	11	
		/	/	[170]
Gastric cancer	Highest vs. Lowest	No data		
	Dose-Response	No data		
**Whole milk (w) and skim milk (s)**				
All-cause mortality	Highest vs. Lowest	1.15 (1.09–1.20) (W)	9	[169]
		/(S)	8	
	Dose-Response	1.10 (1.00–1.21) (W)	6	
		/(S)	6	
Total CVDs mortality	Highest vs. Lowest	1.09 (1.02–1.16) (W)	5	[169]
		/(S)	4	
	Dose-Response	/(W)	4	
		/(S)	4	
CHD	Highest vs. Lowest	No data		
	Dose-Response	No data		
Stroke	Highest vs. Lowest	No data		
	Dose-Response	No data		
HBP	Highest vs. Lowest	No data		
	Dose-Response	No data		
CHF	Highest vs. Lowest	No data		
	Dose-Response	No data		
T2D	Highest vs. Lowest	0.87 (0.78–0.96) (W)	7	[121]
		No data (S)		
	Dose-Response	/(W) /(S)	9 7	[122]
Overweight/Obesity	Highest vs. Lowest	No data		
	Dose-Response	No data		
MetS	Highest vs. Lowest	No data		
	Dose-Response	No data		
Total cancer mortality	Highest vs. Lowest	1.17 (1.08–1.28) (W)	7	[169]
		/(S)	7	
	Dose-Response	1.13 (1.01–1.28) (W)	6	
		/(S)	6	
Total cancer incidence	Highest vs. Lowest	No data		
	Dose-Response	No data		
CRC	Highest vs. Lowest	No data		
	Dose-Response	No data		
Breast cancer	Highest vs. Lowest	/(W)	7 9	[170] [127]
		0.93 (0.84–1.02) (S)	6	[170]
		0.93 (0.85–1.00) (S)	8	[127]
	Dose-Response	/(W)	5	
		0.96 (0.92–1.00) (S)	5	
Gastric cancer	Highest vs. Lowest	No data		
	Dose-Response	No data		
**Yogurt**				
All-cause mortality	Highest vs. Lowest	No data		
	Dose-Response	No data		
Total CVDs mortality	Highest vs. Lowest	No data		
	Dose-Response	No data		
CHD	Highest vs. Lowest	/	5	[171]
	Dose-Response	No data		
Stroke	Highest vs. Lowest	/	5	[171]
	Dose-Response	No data		
HBP	Highest vs. Lowest	No data		
	Dose-Response	No data		
CHF	Highest vs. Lowest	No data		
	Dose-Response	No data		
T2D	Highest vs. Lowest	0.83 (0.70–0.98)	7	[121]
	Dose-Response	0.94 (0.91–0.98)	11	[122]
Overweight/Obesity	Highest vs. Lowest	No data		
	Dose-Response	No data		
MetS	Highest vs. Lowest	0.77 (0.66–0.88)	3	[168]
	Dose-Response	0.82 (0.73–0.91)	3	
Total cancer mortality	Highest vs. Lowest	No data		
	Dose-Response	No data		
Total cancer incidence	Highest vs. Lowest	No data		
	Dose-Response	No data		
CRC	Highest vs. Lowest	No data		
	Dose-Response	No data		
Breast cancer	Highest vs. Lowest	0.91 (0.83–0.99)	7	[170]
		0.90 (0.82–1.00)	5	[127]
	Dose-Response	/	3	
Gastric cancer	Highest vs. Lowest	No data		
	Dose-Response	No data		

^a^ Serving/day compared with no dairy product consumption.

**Table 7 nutrients-14-05115-t007:** Association between total legumes and soybean intake and incidences of some major chronic diseases/mortality of major disease groups (RR, 95% C.I.). The dose-response meta-analysis always refers to an additional serving of a food per day relative to the average intake of that food observed in a cohort study (50 g both for total legumes and soybean).

Health Outcomes	Unit of Intake	RR (95% C.I.)	N° of Prospective Studies	References
**Total Legumes**				
All-cause mortality	Highest vs. Lowest	0.96 (0.94–1.00)	17	[114]
	Dose-Response	0.96 (0.90–1.01)	6	
		0.88 (0.73–1.03)	/	[115]
Total CVDs mortality	Highest vs. Lowest	No data		
	Dose-Response	No data		
CHD	Highest vs. Lowest	0.91 (0.84–0.99)	10	[118]
		0.91 (0.83–1.00)	6	[176]
	Dose-Response	0.86 (0.78–0.94)	5	[177]
		0.96 (0.92–1.01)	8	[118]
		0.88 (0.78–1.03)	/	[115]
Stroke	Highest vs. Lowest	/ /	6 6	[176] [118]
	Dose-Response	/ / /	5 6 /	[177] [118] [115]
HBP	Highest vs. Lowest	0.92 (0.86–0.98)	6	[120]
	Dose-Response	0.98 (0.95–1.01)	5	
CHF	Highest vs. Lowest	No data		
	Dose-Response	No data		
T2D	Highest vs. Lowest	/	13	[176]
	Dose-Response	/ / /	2 / 12	[177] [115] [122]
Overweight/Obesity	Highest vs. Lowest	0.87 (0.81–0.94)	1	[123]
	Dose-Response	0.88 (0.84–0.93)	1	
MetS	Highest vs. Lowest	No data		
	Dose-Response	No data		
Total cancer mortality	Highest vs. Lowest	No data		
	Dose-Response	No data		
Total cancer incidence	Highest vs. Lowest	0.97 (0.93–1.01)		
	Dose-Response	No data		
CRC	Highest vs. Lowest	/	11	[124]
	Dose-Response	/ / /	4 10 /	[125] [124] [115]
Breast cancer	Highest vs. Lowest	No data		
	Dose-Response	No data		
Gastric cancer	Highest vs. Lowest	/	9	[128]
	Dose-Response	No data		
**Soybean**				
All-cause mortality	Highest vs. Lowest	No data		
	Dose-Response	No data		
Total CVDs mortality	Highest vs. Lowest	No data		
	Dose-Response	No data		
CHD	Highest vs. Lowest	/	7	[178]
	Dose-Response	No data		
Stroke	Highest vs. Lowest	/	8	[178]
	Dose-Response	No data		
HBP	Highest vs. Lowest	No data		
	Dose-Response	No data		
CHF	Highest vs. Lowest	No data		
	Dose-Response	No data		
T2D	Highest vs. Lowest	0.87 (0.74–1.01)	7	[121]
	Dose-Response	No data		
Overweight/Obesity	Highest vs. Lowest	No data		
	Dose-Response	No data		
MetS	Highest vs. Lowest	No data		
	Dose-Response	No data		
Total cancer mortality	Highest vs. Lowest	/	10	[179]
	Dose-Response	No data		
Total cancer incidence	Highest vs. Lowest	0.90 (0.83–0.96)	35	[179]
	Dose-Response	No data		
CRC	Highest vs. Lowest	0.88 (0.76–1.02)	4	[179]
	Dose-Response	No data		
Breast cancer	Highest vs. Lowest	0.96 (0.90–1.02)	10	[179]
		0.92 (0.84–1.00)	10	[127]
	Dose-Response	0.91 (0.84–1.00)	7	
		0.89 (0.79–0.99) ^a^	14	[180]
Gastric cancer	Highest vs. Lowest	No data		
	Dose-Response	No data		

^a^ To be referred only to soy isoflavones.

**Table 8 nutrients-14-05115-t008:** Association between nuts and seeds intake and incidences of some major chronic diseases/mortality of major disease groups (RR, 95% C.I.). The dose-response meta-analysis always refers to an additional serving of a food per day relative to the average intake of that food observed in a cohort study (28 g for nuts and seeds).

Health Outcomes	Unit of Intake	RR (95% C.I.)	N° of Prospective Studies	References
**Nuts**				
All-cause mortality	Highest vs. Lowest	0.80 (0.74–0.86)	16	[114]
	Dose-Response	0.76 (0.69–0.84)	16 /	[114] [115]
Total CVDs mortality	Highest vs. Lowest	No data		
	Dose-Response	No data		
CHD	Highest vs. Lowest	0.70 (0.57–0.82)	3	[131]
		0.80 (0.62–1.03)	4	[118]
	Dose-Response	/ /	4 /	[118] [115]
		0.76 (0.69–0.84) ^a^	4	[177]
Stroke	Highest vs. Lowest	/	6	[118]
	Dose-Response	/ ^a^ / /	6 6 /	[177] [118] [115]
HBP	Highest vs. Lowest	0.85 (0.78–0.92)	4	[120]
	Dose-Response	/	4	
CHF	Highest vs. Lowest	/	3	[118]
	Dose-Response	1.09 (0.97–1.22)	2	
T2D	Highest vs. Lowest	No data		
	Dose-Response	0.87 (0.81–0.94) ^a^	2	[177]
		0.79 (0.70–0.90)	/	[115]
		/	7	[122]
Overweight/Obesity	Highest vs. Lowest	0.91 (0.80–1.03)	3	[123]
	Dose-Response	/	3	
		0.93 (0.88–0.98)	5	[190]
MetS	Highest vs. Lowest	No data		
	Dose-Response	No data		
Total cancer mortality	Highest vs. Lowest	No data		
	Dose-Response	No data		
Total cancer incidence	Highest vs. Lowest	0.97 (0.94–1.00)	/	[90]
	Dose-Response	No data		
CRC	Highest vs. Lowest	0.96 (0.90–1.02)	6	[124]
	Dose-Response	/ /	4 /	[124] [115]
Breast cancer	Highest vs. Lowest	/	3	[127]
	Dose-Response	/	3	
Gastric cancer	Highest vs. Lowest	No data		
	Dose-Response	No data		
**Seeds**				
All-cause mortality	Highest vs. Lowest	No data		
	Dose-Response	No data		
Total CVDs mortality	Highest vs. Lowest	No data		
	Dose-Response	No data		
CHD	Highest vs. Lowest	No data		
	Dose-Response	No data		
Stroke	Highest vs. Lowest	No data		
	Dose-Response	No data		
HBP	Highest vs. Lowest	No data		
	Dose-Response	No data		
CHF	Highest vs. Lowest	No data		
	Dose-Response	No data		
T2D	Highest vs. Lowest	No data		
	Dose-Response	No data		
Overweight/Obesity	Highest vs. Lowest	No data		
	Dose-Response	No data		
MetS	Highest vs. Lowest	No data		
	Dose-Response	No data		
Total cancer mortality	Highest vs. Lowest	No data		
	Dose-Response	No data		
Total cancer incidence	Highest vs. Lowest	No data		
	Dose-Response	No data		
CRC	Highest vs. Lowest	No data		
	Dose-Response	No data		
Breast cancer	Highest vs. Lowest	No data		
	Dose-Response	No data		
Gastric cancer	Highest vs. Lowest	No data		
	Dose-Response	No data		

^a^ Based on intake of 4 servings/week of nuts.

**Table 9 nutrients-14-05115-t009:** Association between whole and refined grains (cereal grains) intake and incidences of some major chronic diseases/mortality of major disease groups (RR, 95% C.I.). The dose-response meta-analysis always refers to an additional serving of a food per day relative to the average intake of that food observed in a cohort study (30 g for whole and refined grains).

Health Outcomes	Unit of Intake	RR (95% C.I.)	N° of Prospective Studies	References
**Whole Grains**				
All-cause mortality	Highest vs. Lowest	0.88 (0.84–0.92)	19	[114]
	Dose-Response	0.92 (0.89–0.95)	11 /	[114] [115]
Total CVDs mortality	Highest vs. Lowest	0.79 (0.73–0.85) ^a^	11	[130]
	Dose-Response	No data		
CHD	Highest vs. Lowest	0.81 (0.75–0.86)	7	[131]
		0.85 (0.81–0.90)	7	[118]
	Dose-Response	0.95 (0.92–0.98)	5 /	[118] [115]
Stroke	Highest vs. Lowest	0.83 (0.68–1.02) ^a^	4	[130]
		0.91 (0.82–1.02)	7	[118]
	Dose-Response	0.99 (0.95–1.03)	4 /	[118] [115]
HBP	Highest vs. Lowest	0.86 (0.79–0.93)	4	[120]
	Dose-Response	0.92 (0.87–0.98)	4	
CHF	Highest vs. Lowest	0.91 (0.85–0.97)	5	[118]
	Dose-Response	0.96 (0.95–0.97)	2	
T2D	Highest vs. Lowest	No data		
	Dose-Response	0.79 (0.72–0.87) ^b^	6	[130]
		0.88 (0.83–0.93)	/	[115]
		0.87 (0.82–0.93)	12	[122]
Overweight/Obesity	Highest vs. Lowest	0.85 (0.79–0.91)	5	[123]
	Dose-Response	0.93 (0.89–0.96)	3	
MetS	Highest vs. Lowest	No data		
	Dose-Response	No data		
Total cancer mortality	Highest vs. Lowest	No data		
	Dose-Response	No data		
Total cancer incidence	Highest vs. Lowest	0.93 (0.88–0.98)	/	[90]
	Dose-Response	No data		
CRC	Highest vs. Lowest	0.88 (0.83–0.94)	10	[124]
	Dose-Response	0.83 (0.79–0.89)	6	[125]
		0.95 (0.93–0.97)	2 /	[124] [115]
Breast cancer	Highest vs. Lowest	No data		
	Dose-Response	No data		
Gastric cancer	Highest vs. Lowest	No data		
	Dose-Response	0.83 (0.78–0.89) ^c^	6	[196]
**Refined Grains**				
All-cause mortality	Highest vs. Lowest	/	4	[114]
	Dose-Response	0.99 (0.97–1.01)	4 /	[114] [115]
Total CVDs mortality	Highest vs. Lowest	No data		
	Dose-Response	No data		
CHD	Highest vs. Lowest	1.11 (0.99–1.25)	5	[118]
	Dose-Response	1.01 (0.99–1.04)	4 /	[118] [115]
Stroke	Highest vs. Lowest	/	6	[118]
	Dose-Response	/ /	4 /	[118] [115]
HBP	Highest vs. Lowest	0.95 (0.88–1.03)	3	[120]
	Dose-Response	/	3	
CHF	Highest vs. Lowest	/	1	[118]
	Dose-Response	/	1	
T2D	Highest vs. Lowest	No data		
	Dose-Response	1.01 (1.00–1.03)	14	[122]
		0.98 (0.96–1.01)	/	[115]
Overweight/Obesity	Highest vs. Lowest	/	3	[123]
	Dose-Response	1.05 (1.00–1.10)	3	
MetS	Highest vs. Lowest	No data		
	Dose-Response	No data		
Total cancer mortality	Highest vs. Lowest	No data		
	Dose-Response	No data		
Total cancer incidence	Highest vs. Lowest	No data		
	Dose-Response	No data		
CRC	Highest vs. Lowest	/	9	[124]
	Dose-Response	No data		
Breast cancer	Highest vs. Lowest	No data		
	Dose-Response	No data		
Gastric cancer	Highest vs. Lowest	No data		
	Dose-Response	No data		

^a^ 2.5 vs. 0.2 servings/day; ^b^ each, 2 servings/day; ^c^ for an increment of three servings daily.

**Table 10 nutrients-14-05115-t010:** Association between intake of animal- and plant-based protein sources and incidences of some major chronic diseases (RR, 95% C.I.)/mortality of major disease groups (HR, 95% C.I.).

Health Outcomes	Unit of Intake	HR/RR (95% C.I.)	N° of Prospective Studies	References
**Animal Protein Sources**				
All-cause mortality	Highest vs. Lowest	/ ^a^ / ^b^	2 11	[199] [200]
	Dose-Response	No data		
Total CVDs mortality	Highest vs. Lowest	1.08 (1.01–1.16) ^a^	2	[199]
		/ ^b^	8	[200]
	Dose-Response	No data		
CHD	Highest vs. Lowest	/ ^c^	5	[201]
	Dose-Response	No data		
Stroke	Highest vs. Lowest	No data		
	Dose-Response	No data		
HBP	Highest vs. Lowest	/ ^c^	5	[202]
	Dose-Response	No data		
CHF	Highest vs. Lowest	No data		
	Dose-Response	No data		
T2D	Highest vs. Lowest	1.13 (1.06–1.21) ^c^	3	[201]
		1.14 (1.09–1.19) ^c^	9	[121]
	Dose-Response	1.12 (1.08–1.17) ^c^	8	[122]
Overweight/Obesity	Highest vs. Lowest	No data		
	Dose-Response	No data		
MetS	Highest vs. Lowest	No data		
	Dose-Response	No data		
Total cancer mortality	Highest vs. Lowest	/ ^a^ / ^b^	2 9	[199] [200]
	Dose-Response	No data		
Total cancer incidence	Highest vs. Lowest	No data		
	Dose-Response	No data		
CRC	Highest vs. Lowest	No data		
	Dose-Response	No data		
Breast cancer	Highest vs. Lowest	No data		
	Dose-Response	No data		
Gastric cancer	Highest vs. Lowest	No data		
	Dose-Response	No data		
**Plant-Based Protein Sources**				
All-cause mortality	Highest vs. Lowest	0.90 (0.86–0.95) ^d^	2	[199]
		0.92 (0.87–0.97) ^b^	13	[200]
	Dose-Response	No data		
Total CVDs mortality	Highest vs. Lowest	0.88 (0.80–0.97) ^d^	2	[199]
		0.88 (0.80–0.96) ^b^	10	[200]
	Dose-Response	No data		
CHD	Highest vs. Lowest	0.91 (0.80–1.02) ^c^	4	[202]
	Dose-Response	No data		
Stroke	Highest vs. Lowest	No data		
	Dose-Response	No data		
HBP	Highest vs. Lowest	0.87 (0.74–1.01) ^c^	5	[202]
	Dose-Response	No data		
CHF	Highest vs. Lowest	No data		
	Dose-Response	No data		
T2D	Highest vs. Lowest	0.91 (0.84–0.98) ^c^	3	[201]
		/ ^c^	9	[121]
	Dose-Response	0.87 (0.74–1.01) ^c^	8	[122]
Overweight/Obesity	Highest vs. Lowest	No data		
	Dose-Response	No data		
MetS	Highest vs. Lowest	No data		
	Dose-Response	No data		
Total cancer mortality	Highest vs. Lowest	/ ^d^ / ^b^	2 9	[199] [200]
	Dose-Response	No data		
Total cancer incidence	Highest vs. Lowest	No data		
	Dose-Response	No data		
CRC	Highest vs. Lowest	No data		
	Dose-Response	No data		
Breast cancer	Highest vs. Lowest	No data		
	Dose-Response	No data		
Gastric cancer	Highest vs. Lowest	No data		
	Dose-Response	No data		

^a^ Highest vs. lowest intake category of animal protein sources measured as % of total diet energy (thus “per 10% energy increment”); ^b^ highest vs. lowest intake of animal protein sources through the computation of estimates using the fixed effects model and Orsini method; ^c^ intake increment per 5% energy/day; ^d^ highest vs. lowest intake category of plant-based protein sources measured as % of total diet energy (thus “per 3% energy increment”).

## Data Availability

Not applicable.

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
