# Peer review of "Animal- and Plant-Based Protein Sources: A Scoping Review of Human Health Outcomes and Environmental Impact"

_nutrients, 2022, doi:10.3390/nu14235115_

Round 1

Reviewer 1 Report

The authors describe a review of the impact of diet on disease, death, and even the environment. This article provides very interesting and important information for the readers, but the following points are required for the correction.

Major points.

1. Since the effects of lipids, fiber, etc. are described in addition to protein, protein alone is misleading and should be changed in the title

2. Provide definitions of Highest and Lowest in Table 2 and Table 10

3. Add a detailed explanation of Table 2

4. Describe how MDS is calculated

Minor points:

1. Table 12 is missing

Author Response

Thank you very much for your excellent critique of our manuscript.

Major points.

  1. Since the effects of lipids, fiber, etc. are described in addition to protein, protein alone is misleading and should be changed in the title

Ths is the only point we disagree with. We mention but do not discuss fiber, lipid, etc. as outlined in the Methods section. We prefer to keep the title as is.

  1. Provide definitions of Highest and Lowest in Table 2 and Table 10

This is all in Ref 91, where the MDS is also calculated.

  1. Add a detailed explanation of Table 2

Done

  1. Describe how MDS is calculated

Readers are referred to Ref. 91.

Minor points:

  1. Table 12 is missing

Thank you for pointing this out, that was our mistake as there was no Table 12 to start with. We deleted it from the text and we thank you once more.

Reviewer 2 Report

This review article is devoted to the study of protein sources of animal and plant origin and shows the main aspects of the impact on both human health and the environment. The review is well written and is of great interest to the readers of the journal Nutrients. These comprehensive data have great potential, however, my comments are minor:

(1) Why did the authors not use such a large Scifinder database, what explains the use of the presented databases?

(2) What is the meaning of the links highlighted in tables 2,3,4,5,6,7,8 of different colors? Correct it properly or explain what it means.

(3) What is the meaning of the void in tables 2,3,4,5,6,7,8, what information do they carry? Remove unnecessary data! Please fix it!

(4) Check the correctness of the list of references, there are inaccuracies.

Author Response

First of all, thank you for taking the time to critically read our huge manuscript and for providing such useful insight.

(1) Why did the authors not use such a large Scifinder database, what explains the use of the presented databases?

We checked Scifinder (the old CAS), but it mostly applies to chemical compounds and left many of our references out. It would not add anything to the paper.

(2) What is the meaning of the links highlighted in tables 2,3,4,5,6,7,8 of different colors? Correct it properly or explain what it means.

That was a software glitch that we now corrected. Thank you.

(3) What is the meaning of the void in tables 2,3,4,5,6,7,8, what information do they carry? Remove unnecessary data! Please fix it!

To make Tables more consistent we now added: "No data" where data were not available.

(4) Check the correctness of the list of references, there are inaccuracies.

We triple-checked the references and believe they are now perfect. Thank you for going through them!